# Generative Modelling of Stochastic Actions with Arbitrary Constraints in Reinforcement Learning

**Chen Changyu[1], Ramesha Karunasena[1], Thanh Hong Nguyen[2],**
**Arunesh Sinha[3], Pradeep Varakantham[1]**
Singapore Management University[1], University of Oregon[2], Rutgers University[3]
cychen.2020@phdcs.smu.edu.sg, rameshak@smu.edu.sg,
thanhhng@cs.uoregon.edu, arunesh.sinha@rutgers.edu, pradeepv@smu.edu.sg

## Abstract

Many problems in Reinforcement Learning (RL) seek an optimal policy with large discrete multidimensional yet unordered action spaces; these include problems in randomized allocation of resources such as placements of multiple security resources and emergency response units, etc. A challenge in this setting is that the underlying action space is categorical (discrete and unordered) and large, for which existing RL methods do not perform well. Moreover, these problems require validity of the realized action (allocation); this validity constraint is often difficult to express compactly in a closed mathematical form. The allocation nature of the problem also prefers stochastic optimal policies, if one exists. In this work, we address these challenges by (1) applying a (state) conditional normalizing flow to compactly represent the stochastic policy — the compactness arises due to the network only producing one sampled action and the corresponding log probability of the action, which is then used by an actor-critic method; and (2) employing an invalid action rejection method (via a valid action oracle) to update the base policy. The action rejection is enabled by a modified policy gradient that we derive. Finally, we conduct extensive experiments to show the scalability of our approach compared to prior methods and the ability to enforce arbitrary state-conditional constraints on the support of the distribution of actions in any state[1].

## 1 Introduction

Adaptive resource allocation problems with multiple resource types (e.g., fire trucks, ambulances, police vehicles) are ubiquitous in the real world [13, 23, 32]. One example is allocating security resources and emergency response vehicles to different areas depending on incidents [23]. There are many other similar problems in aggregation systems for mobility/transportation, logistics etc [13]. In this paper, we are interested in addressing the multiple difficult challenges present in such adaptive resource allocation problems: (a) Combinatorial action space, as the number of resource allocations is combinatorial; (b) *Categorical* action space, as there is no ordering of resource allocations with respect to the overall objective (as increasing or decreasing different resource types at different locations can have different impact on overall objective) ; (c) Constraints on feasible allocations and switching between allocations; and (d) Finally, uncertainty in demand for resources.

Existing research in RL for constrained action spaces has considered resource allocation problems with a single type of resource, thereby introducing order in action space [1]. In such ordered action spaces, actions can be converted into continuous space and this allows for the usage of continuous action RL methods (e.g., DDPG). In problems of interest in this paper, we are interested in problems

---

[1]Our implementation is available at https://github.com/cameron-chen/flow-iar.

37th Conference on Neural Information Processing Systems (NeurIPS 2023).

with multiple resource types (e.g., fire truck, ambulance, police). These problems have a large action space that is discrete and unordered (categorical) and there are constraints on feasible allocations (e.g., no two police vehicles can be more than 3 km away, cannot move vehicles too far away from time step to time step). Furthermore, such constraints are easy to validate by a validity oracle given any allocation action, but are hard to represent as mathematical constraints on the support of the distribution of actions (in each state) as they often require exponentially many inequalities [4, 35]. An important consideration in allocation problems is randomized (stochastic) allocation arising from issues of fair division of indivisible resources so that an allottee is not starved of resources forever [4]. Thus, we aim to output stochastic optimal policies, if one exists.

Towards addressing such resource allocation problems at scale, we propose to employ generative policies in RL. Specifically, we propose a new approach that incorporates discrete normalizing flow policy in an actor-critic framework to explore and learn in the aforementioned constrained, categorical and adaptive resource allocation problems. Prior RL approaches for large discrete multidimensional action space include ones that assume a factored action space with independent dimensions, which we call as the factored or marginal approach, since independence implies that any joint distribution over actions can be represented as the product of marginal distributions over each action dimension. Other approaches convert the selection of actions in multiple dimensions into a sequential selection approach. Both these approaches are fundamentally limited in expressivity, which we reveal in detail in our experiments. Next, we provide a formal description of the problem that we tackle.

**Problem Statement:** A Markov Decision Process (MDP) is represented by the tuple $\langle \mathcal{S}, \mathcal{A}, P, r, \gamma, b_0 \rangle$, where an agent can be in any state $s_t \in \mathcal{S}$ at a given time $t$. The agent takes an action $a_t \in \mathcal{A}$, causing the environment to transition to a new state $s_{t+1}$ with a probability $P : \mathcal{S} \times \mathcal{A} \times \mathcal{S} \mapsto [0, 1]$. Subsequently, the agent receives a reward $r : \mathcal{S} \times \mathcal{A} \mapsto \mathbb{R}$. In the infinite-horizon setting, the discounted factor is $0 < \gamma < 1$. The distribution of the initial state is $b_0$.

In our work, we focus on a categorical action space, $\mathcal{A}$. Categorical action spaces consist of discrete, unordered actions with no inherent numerical relationship between them. We assume that for any state $s$, there is a set of valid actions, denoted by $\mathcal{C}(s) \subseteq \mathcal{A}$. There is an oracle to answer whether an action $a \in \mathcal{C}(s)$, but the complex constraint over categorical space cannot be expressed succinctly using closed form mathematical formula. Note that the constraint is *not* the same for every state. Our objective is to learn a stochastic policy, $\pi(\cdot|s)$, which generates a *probability distribution over actions for state $s$ with support only over the valid actions $\mathcal{C}(s)$ in state $s$*. We call the set of such stochastic policies as valid policies $\Pi^{\mathcal{C}}$ given the per state constraint $\mathcal{C}$. We aim to maximize the long-term expected rewards over valid policies, that is, $\max_{\pi \in \Pi^c} J(\pi)$, where $J$ is as follows:

$$J(\pi) = \mathbb{E}_{s \sim b_0}[V(s; \pi)] \text{ where } V(s; \pi) = \mathbb{E}\left[\sum_{t=0}^{\infty} \gamma^t r(s_t, a_t) | s_0 = s; \pi\right] \tag{1}$$

In addition, we also consider settings with *partial observability* where the agent observes $o_t \in \mathcal{O}$ at time $t$ where the observation arises from the state $s_t$ with probability $O : \mathcal{O} \times \mathcal{S} \mapsto [0, 1]$. In this case, the optimal policy is a stochastic policy, $\pi(\cdot|h)$, where $h$ is the history of observations till current time. For partial observability, we consider an unconstrained setting, as the lack of knowledge of the true state results in uncertainty about which constraint to enforce, which diverges from the focus in this work but is an interesting future research direction to explore. Thus, with partial observability, we search over stochastic policies $\Pi$ that maximize the long term return, that is, $\max_{\pi \in \Pi} J(\pi)$. $J(\pi)$ is the same as stated in Equation 1 but where the expectation is also over the observation probability distribution in addition to the standard transition and stochastic policy probability distributions.

**Contribution:** We propose two key innovations to address the problem above. *First*, we present a conditional Normalizing Flow-based [26] Policy Network, which leverages Argmax Flow [16] to create a minimal representation of the policy for policy gradient algorithms. To the best of our knowledge, this is the first use of discrete normalizing flow in RL. *Second*, we demonstrate how to train the flow policies within the A2C framework. In particular, we need an estimate of the log probability of the action sampled from the stochastic policy but Argmax Flow provides only a biased lower bound via the evidence lower bound (ELBO). Thus, we design an effective *sandwich estimator* for the log probability that is sandwiched between the ELBO lower bound and an upper bound based on $\chi^2$ divergence. *Third*, we propose a policy gradient approach which is able to reject invalid actions (that do not satisfy constraints) referred to as Invalid Action Rejection Advantage Actor-Critic (IAR-A2C). IAR-A2C queries the constraint oracle and ensures validity of actions in every state (in fully observable setting) by rejecting all invalid actions. We derive a new policy gradient estimator

for IAR-A2C. Figure 1 provides an overview of our architecture. Finally, our extensive experimental results reveal that our approach outperforms prior baselines in different environments and settings.

## 2 Background

**Normalizing Flows:** Normalizing flows are a family of generative models that can provide both efficient sampling and density estimation. Their main idea is to construct a series of invertible and differentiable mappings that allow transforming a simple probability distribution into a more complex one. Given $\mathcal{V} = \mathcal{Z} = \mathbb{R}^d$ with densities $p_V$ and $p_Z$ respectively, normalizing flows [26] aim to learn a bijective and differentiable transformation $f : \mathcal{Z} \to \mathcal{V}$. This deterministic transformation allows us to evaluate the density at any point $v \in \mathcal{V}$ based on the density of $z \in \mathcal{Z}$, as follows:

$$p_V(\boldsymbol{v}) = p_Z(\boldsymbol{z}) \cdot \left| \det \frac{\mathrm{d}\boldsymbol{z}}{\mathrm{d}\boldsymbol{v}} \right|, \quad \boldsymbol{v} = f(\boldsymbol{z}) \tag{2}$$

In this context, $p_Z$ can be any density, though it is typically chosen as a standard Gaussian and $f$ is represented by a neural network. Consequently, normalizing flows offer a powerful tractable framework for learning complex density functions. However, the density estimation presented in Equation 2 is limited to continuous probability distributions. To enable the learning of probability mass functions ($P$) on categorical discrete data, such as natural language, Argmax Flow [16] proposed to apply the argmax operation on the output of continuous flows. Let's consider $\boldsymbol{v} \in \mathbb{R}^{D \times M}$ and $\boldsymbol{x} \in \{1, \ldots, M\}^D$. The argmax operation is interpreted as a surjective flow layer $\boldsymbol{v} \mapsto \boldsymbol{x}$, which is deterministic in one direction ($x_d = \arg\max_m \boldsymbol{v}_d$, written compactly as $\boldsymbol{x} = \arg\max \boldsymbol{v}$) and stochastic in the other ($\boldsymbol{v} \sim q(\cdot|\boldsymbol{x})$). With this interpretation, the argmax operation can be considered a probabilistic right-inverse in the latent variable model expressed by:

$$P(\boldsymbol{x}) = \int P(\boldsymbol{x}|\boldsymbol{v})p(\boldsymbol{v})\mathrm{d}\boldsymbol{v}, \quad P(\boldsymbol{x}|\boldsymbol{v}) = \delta(\boldsymbol{x} = \mathrm{argmax}(\boldsymbol{v})) \tag{3}$$

where $\mathrm{argmax}$ is applied in the last dimension of $\boldsymbol{v}$. In this scenario, the density model $p(\boldsymbol{v})$ is modeled using a normalizing flow. The learning process involves introducing a variational distribution $q(\boldsymbol{v}|\boldsymbol{x})$, which models the probabilistic right-inverse for the argmax surjection, and optimizing the evidence lower bound (ELBO), which is the RHS of the following inequality:

$$\log P(\boldsymbol{x}) \geq \mathbb{E}_{\boldsymbol{v} \sim q(\cdot|\boldsymbol{x})}[\log P(\boldsymbol{x}|\boldsymbol{v}) + \log p(\boldsymbol{v}) - \log q(\boldsymbol{v}|\boldsymbol{x})] = \mathbb{E}_{\boldsymbol{v} \sim q(\cdot|\boldsymbol{x})}[\log p(\boldsymbol{v}) - \log q(\boldsymbol{v}|\boldsymbol{x})] = \mathcal{L}$$

The last equality holds under the constraint that the support of $q(\boldsymbol{v}|\boldsymbol{x})$ is enforced to be only over the region $\mathcal{S} = \{\boldsymbol{v} \in \mathbb{R}^{D \times M} : \boldsymbol{x} = \arg\max \boldsymbol{v}\}$ which ensures that $P(\boldsymbol{x}|\boldsymbol{v}) = 1$. From standard variational inference results, $\log P(\boldsymbol{x}) - \mathcal{L} = \mathrm{KL}(q(\boldsymbol{v}|\boldsymbol{x})||p(\boldsymbol{v}|\boldsymbol{x}))$, which also approaches 0 as the approximate posterior $q(\boldsymbol{v}|\boldsymbol{x})$ comes closer to the true posterior $p(\boldsymbol{v}|\boldsymbol{x})$ over the training time.

$\chi^2$ **Upper Bound:** Variational inference involves proposing a family of approximating distributions and finding the family member that is closest to the posterior. Typically, the Kullback-Leibler (KL) divergence $\mathrm{KL}(q||p)$ is employed to measure closeness, where $q(\boldsymbol{v}|\boldsymbol{x})$ represents a variational family. This approach yields ELBO of the evidence $\log P(\boldsymbol{x})$ as described above.

Instead of using KL divergence, the authors in [8] suggest an alternative of $\chi^2$-divergence to measure the closeness. As a result, they derived an upper bound of the evidence, known as CUBO: $\log P(\boldsymbol{x}) \leq \frac{1}{2} \log \mathbb{E}_{\boldsymbol{v} \sim q(\cdot|\boldsymbol{x})} \left[ \left( \frac{p(\boldsymbol{x},\boldsymbol{v})}{q(\boldsymbol{v}|\boldsymbol{x})} \right)^2 \right]$. Similar to the ELBO in Argmax Flow, CUBO can be further simplified under the constraint that the support of $q(\cdot|\boldsymbol{x})$ is restricted to the region $\mathcal{S}$:

$$\log P(\boldsymbol{x}) \leq \frac{1}{2} \log \mathbb{E}_{\boldsymbol{v} \sim q(\cdot|\boldsymbol{x})} \left[ \left( \frac{p(\boldsymbol{v})}{q(\boldsymbol{v}|\boldsymbol{x})} \right)^2 \right] = \mathcal{L}_{\chi^2} \tag{4}$$

Also, $\mathcal{L}_{\chi^2} - \log P(\boldsymbol{x}) = \frac{1}{2} \log(1 + D_{\chi^2}(p(\boldsymbol{v}|\boldsymbol{x}))||q(\boldsymbol{v}|\boldsymbol{x})))$, hence the gap between the ELBO and CUBO approaches 0 as the approximate posterior $q(\cdot|\boldsymbol{x})$ becomes closer to the true posterior $p(\cdot|\boldsymbol{x})$.

## 3 Flow-based Policy Gradient Algorithm with Invalid Action Rejection

We first present the Flow-based Policy Network, which leverages Argmax Flow to create a minimal representation of the policy for policy gradient algorithms, and then construct flow policies within

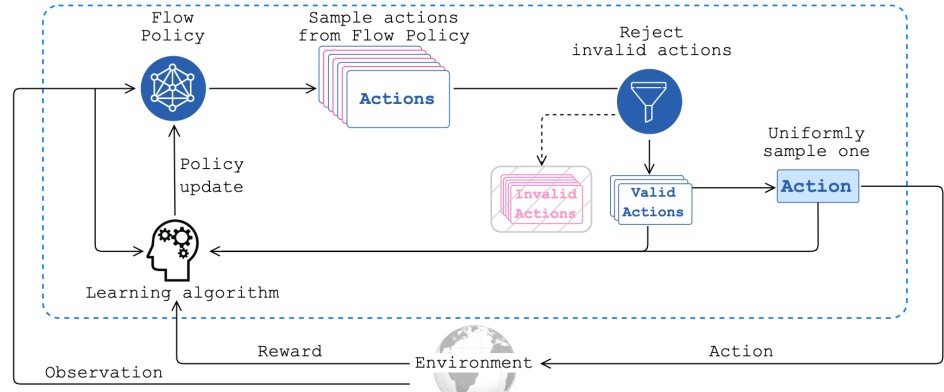

Figure 1: **Our IAR-A2C framework**. At each time step, an initial batch of action samples, along with their log probabilities, are generated using the Flow Policy. Invalid actions from this batch are rejected using an oracle. A single action is then uniformly sampled from the remaining valid ones, and executed. This selected action and the valid action set are stored along with the resulting state and reward. This collective experience is subsequently utilized to update the Flow Policy.

the A2C framework. Our exposition will present policies conditioned on state, but the framework works for partial observability also by using the sequence of past observations as state. After this, for the fully observable setting only, we introduce our novel policy gradient algorithm called IAR-A2C that enforces state dependent constraints.

### 3.1 Flow-based Policy Network

In policy gradient algorithms for categorical actions, the standard approach is to model the entire policy, denoted as $\pi(\cdot|s)$, which allows us to sample actions and obtain their corresponding probabilities. The policy is parameterized using a neural network where the network generates logits for each action, and then converts these logits into probabilities. The size of the output layer matches that of the action space, which can be prohibitively large for resource allocation problems of interest in this paper. We posit that we can have a better policy representation, as it is sufficient to require samples from the support set of the policy, represented by $a^{(i)} \sim \pi(\cdot|s)$ and probability value $\pi(a^{(i)}|s) > 0$.

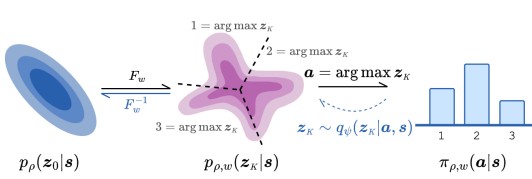

Figure 2: Composition of a conditional flow $p_{\rho,w}(\boldsymbol{z}_K|\boldsymbol{s})$ and argmax transformation resulting in the policy $\pi_{\rho,w}(\boldsymbol{a}|\boldsymbol{s})$. The flow maps from a base distribution $p_{\rho}(\boldsymbol{z}_0|\boldsymbol{s})$ by using a bijection $F_w$. The diagram is adapted from Hoogeboom et al. [16].

Based on this observation, our first contribution is a compact policy representation using Argmax Flow [16], which we refer to as the *Flow Policy*. Argmax Flow is a state-of-the-art discrete normalizing flow model and it has shown great capability of learning categorical data such as in sentence generation. In our context, Flow Policy will output the action (a sample of the flow policy) and its probability, instead of explicitly outputting the entire distribution and sampling from it as in prior work. Once trained, we can sample from the Argmax Flow and, more importantly, estimate the probability of the sample. A normalizing flow model transforms a base distribution, given by a random variable $z_0$, to the desired distribution. In our approach, the desired distribution is the distribution given by policy $\pi$. We adapt Argmax Flow approach for learning a stochastic policy.

Before diving into the specifics of the Flow Policy, we first discuss our rationale for choosing the normalizing flow model over other contemporary deep generative models, such as Generative Adversarial Networks (GANs) [12] and Diffusion Models [15, 29]. GANs, categorized as implicit generative models, do not allow for an estimation of the data's log probability. While prior research has successfully devised an approach for exact log probability estimation with Diffusion Models, these

**Algorithm 1:** ELBO Optimization

1  **Input**: Invertible flow $F_w = f_{w,k} \circ \ldots \circ f_{w,1}$, State encoder $E_\rho$, Posterior $q_\psi$, rollout $\tau$

2  Sample $States = \{(s^{(i)})\}_{i=1}^{B} \sim \tau$

3  **for** $s \in States$ **do**

4     **for** $j \leftarrow 1 : n\_elbo\_steps$ **do**

5        $z^{(j)} = F_w(z_0^{(j)})$, where $z_0^{(j)} \sim E_\rho(s)$

6        $a^{(j)} = \operatorname{argmax} z^{(j)}$

7     **for** $j \leftarrow 1 : n\_elbo\_steps$ **do**

8        $z'^{(jn)} = \operatorname{threshold}_T(u^{(jn)})$, where $\{(u^{(jn)})\}_{n=1}^{N} \sim q_\psi(\cdot|a^{(j)}, s)$

9        $z_0'^{(jn)} = F_w^{-1}(z'^{(jn)})$

10      Take a gradient ascending step of ELBO w.r.t. $\rho, w$ and $\psi$

11      $\mathcal{L} = \frac{1}{N} \sum_{i=1}^{N} \left( \log p_\rho(z_0'^{(jn)}|s) - \sum_{k=1}^{K} \log \left| \det \frac{\partial f_{w,k}}{\partial z_{k-1}'^{(jn)}} \right| - \log q_\psi(z'^{(jn)}|a^{(j)}, s) \right)$

models encounter the problem of slow sampling and expensive probability estimation, necessitating the resolution of an ordinary differential equation [29]. In contrast, normalizing flow has low cost for sampling and probability evaluation in our case, as we find the simple flow model is sufficient to learn a good policy. So normalizing flow is a natural choice to construct our policy.

We demonstrate how to construct the flow policy within the A2C framework. An illustration is provided in Figure 2. To define the policy, $\pi(a|s)$, we first encode the state that is then used to define the base distribution $z_0$. In accordance with standard practice, we select $z_0$ as a Gaussian distribution with parameters $\mu, \sigma$ defined by the state encoder (a state-dependent neural network): $z_0 = \mu_\rho(s) + \sigma_\rho(s) \cdot \epsilon$ where $\epsilon \sim \mathcal{N}(0, 1)$. We write this as $z_0 \sim E_\rho(s)$, where $\rho$ denotes the weights of the state encoder. We then apply a series of invertible transformations given as functions $f_k$ that define the flow, followed by the argmax operation (as defined in background on Argmax flow). Consequently, the final sampled action is given by $a = \operatorname{argmax}(f_K \circ \ldots \circ f_1(z_0))$. Each $f_k$ is an invertible neural network [26] and we use $F_w = f_{w,K} \circ \ldots \circ f_{w,1}$ to denote the composed function, where $w, k$ is the weight of the network representing $f_k$. Thus, sampled action $a = \operatorname{argmax}(F_w(z_0))$ for $z_0 \sim E_\rho(s)$. We use shorthand $\theta = (\rho, w)$ when needed. Also, we use $z_k = f_{w,k}(z_{k-1})$ to denote the output of $f_{w,k}$ and $p_\rho(z_0|s)$ to denote the probability density of $z_0 \sim \mathcal{N}(\mu_\rho, \sigma_\rho)$.

We use a variational distribution $q_\psi$ for the reverse mapping from a discrete action (conditional on state $s$) to a continuous output of $F_w(z_0)$. The corresponding estimate of the log probability of the sampled action $a$, denoted by $\hat{l}_\pi$, is the evidence lower bound ELBO, computed as follows:

$$\hat{l}_\pi(a|s) = \mathcal{L} = \mathbb{E}_{z_K \sim q_\psi(\cdot|a,s)} \left[ \log p_\rho(z_0|s) - \sum_{k=1}^{K} \left( \log \left| \det \frac{\partial f_k}{\partial z_{k-1}} \right| \right) - \log q_\psi(z_K|a,s) \right] \quad (5)$$

To ensure that our approximation closely approximates the evidence, we optimize ELBO progressively, following the training scheme of Argmax flow, as shown in the subroutine Algorithm 1 used within the outer A2C framework in Algorithm 2. The target distribution in this subroutine is given by actions $a^{(j)}$ sampled from the current policy (line 6); thus, this subroutine aims to update the flow network components $(\rho, w)$ to make the distribution of overall policy be closer to this target distribution and to improve the posterior $q_\psi$ estimate (ELBO update in line 11), which gets used in estimating the log probability of action required by the outer A2C framework (shown later in Algorithm 2). The various quantities needed for ELBO update are obtained in lines 8, 9 by passing $a^{(j)}$ back through the invertible flow layers (using $q_\psi$ for the discrete to continuous inverse map).

## 3.2  Policy Gradient with Invalid Action Rejection

We aim to use the flow-based policy within an A2C framework, illustrated in Figure 1; here we describe the same along with how to enforce arbitrary state based constraint on actions. We propose a sandwich estimator which combines ELBO and CUBO to obtain a low-bias estimation of $\log \pi(a|s)$, thus improving the convergence of policy gradient learning. Moreover, to tackle the challenge of aforementioned complex constraints, our algorithm proceeds by sampling a number of actions from

the flow-based policy and then utilizing the constraint oracle to filter the invalid actions. We then provide theoretical results showing the adaptation of the policy gradient computation accordingly.

**Sandwich estimator:** The ELBO $\hat{l}_\pi$ is a biased estimator of $\log \pi(a|s)$, but it is known in literature [5] that $\hat{l}_\pi$ is a consistent estimator (i.e., converges to $\log \pi(a|s)$ in the limit) and there are generative methods based on this consistency property, such as [33]. However, we aim to use $\hat{l}_\pi$ in the policy gradient and stochastic gradient descent typically uses an unbiased gradient estimate (but not always, see [6]). Therefore, we propose to use a new technique which combines ELBO and CUBO to reduce the bias, improving the convergence of our policy gradient based learning process. In particular, we estimate an upper bound of $\log \pi(a|s)$ using the following CUBO:

$$\hat{l}^u_\pi(a|s) = \mathcal{L}_{\chi^2} = \frac{1}{2} \log \mathbb{E}_{z_K \sim q_\psi(z_K|a,s)} \left[ \left( \frac{p_\rho(z_0|s)}{\prod_{k=1}^K |\det \frac{\partial f_{w,k}}{\partial z_{k-1}}| q_\psi(z_K|a,s)} \right)^2 \right] \tag{6}$$

We then use a weighted average of the upper and lower bounds as a low-bias estimate of $\log \pi(a|s)$, denoted by $\widehat{\log p}_{\theta,\psi} = \alpha \hat{l}_\pi + (1-\alpha)\hat{l}^u_\pi$ where $\alpha$ is a hyperparameter. We call this the *sandwich estimator* of log probability. We observed in our experiments that an adaptive $\alpha(\hat{l}_\pi, \hat{l}^u_\pi)$ as a function of the two bounds provides better results than a static $\alpha = \frac{1}{2}$ (see Appendix B.1 for more details).

**Constraints:** Since the agent only has oracle access to the constraints, existing safe exploration approaches [20, 24] are not directly applicable to this setting. If the agent queries the validity of all actions, then it gains complete knowledge of the valid action set $\mathcal{C}(s)$. However, querying action validity for the entire action space at each timestep can be time-consuming, particularly when the action space is large, e.g., 1000 categories. We demonstrate this issue in our experiments.

To address this challenge of complex *state-dependent* constraints in the full observation setting, we propose a new policy gradient algorithm called Invalid Action Rejection Advantage Actor-Critic (IAR-A2C). IAR-A2C operates by sampling a set of actions from the current flow-based policy and then leveraging the constraint oracle to reject all invalid actions, enabling the agent to explore safely with only valid actions. We then derive the policy gradient estimator for this algorithm. Recall that $\mathcal{C}(s)$ is the set of valid actions in state $s$ and let $\mathcal{C}^I(s)$ be the set of invalid actions. We sample an action from $\pi_\theta(a|s)$ (recall $\theta = (\rho, w)$ when using Flow Policy), rejecting any invalid action till a valid action is obtained. Clearly, the effective policy $\pi'$ induced by this rejection produces only valid actions. In fact, by renormalizing we obtain $\pi'_\theta(a|s) = \frac{\pi_\theta(a|s)}{\sum_{a_i \in \mathcal{C}(s)} \pi_\theta(a_i|s)}$ for $a \in \mathcal{C}(s)$. For the purpose of policy gradient, we need to obtain the gradient of the long term reward with $\pi'$: $J(\theta) = \mathbb{E}_{s \sim b_0}[V(s; \pi')]$. We show that:

**Theorem 1** *With $\pi'$ defined as above,*

$$\nabla_\theta J(\theta) = \mathbb{E}_{\pi'} \left[ Q^{\pi'}(s,a) \nabla_\theta \log \pi_\theta(a|s) - Q^{\pi'}(s,a) \frac{\sum_{a \in \mathcal{C}(s)} \nabla_\theta \pi_\theta(a|s)}{\sum_{a \in \mathcal{C}(s)} \pi_\theta(a|s)} \right] \tag{7}$$

In practice, analogous to standard approach in literature [22] to reduce variance, we use the TD error (an approximation of advantage) form instead of the Q function in Theorem 1. Then, given the network $\pi_\theta$, the empirical estimate of the first gradient term is readily obtainable from trajectory samples of $\pi'$. To estimate the second term, for every state $s$ in the trajectories, we sample $S$ actions from $\pi_\theta(a|s)$ and reject any $a \in \mathcal{C}^I(s)$ to get $l \leq S$ valid actions $a_1, ..., a_l$. We then apply Lemma 1:

**Lemma 1** $\frac{1}{l} \sum_{j \in [l]} \nabla_\theta \log \pi_\theta(a_j|s)$ *and* $\frac{l}{S}$ *are unbiased estimates of* $\sum_{a \in \mathcal{C}(s)} \nabla_\theta \pi_\theta(a|s)$ *and* $\sum_{a \in \mathcal{C}(s)} \pi_\theta(a|s)$ *respectively.*

The full approach is shown in Algorithm 2, with the changes from standard A2C highlighted in red. Recall that, $\theta = (\rho, w)$ is the parameter of the flow-based policy network, $\phi$ is the parameter of the critic, and $\psi$ is the parameter of the variational distribution network $q_\psi$ for ELBO.

## 4 Related Work

**Large discrete action space.** Dulac-Arnold et al. [10] attempt to solve this problem by embedding discrete actions into a continuous space; however, this approach does not work for our unordered

**Algorithm 2:** IAR-A2C

---

**1** Initialize step count $t \leftarrow 1$, Initialize episode counter $E \leftarrow 1$
**2 repeat**
**3**     Reset gradients: $d(\theta, \psi) \leftarrow 0$, $d\phi \leftarrow 0$, Initialize experience set $\tau$, $t_{start} = t$, get state $s_t$
**4**     **repeat**
**5**        Execute $a_t$ according to policy $\pi'_\theta(a_t|s_t)$   *// Defined in Section 3.2*
**6**        Get a number of valid and sampled actions $l_t$ and $S_t$
**7**        Receive reward $r_t$ and new state $s_{t+1}$
**8**        Add new experience $\tau \leftarrow \tau \cup \{s_t, a_t, s_{t+1}, r_t\}$, and update $t \leftarrow t + 1$
**9**     **until** terminal $s_t$ **or** $t - t_{start} = t_{max}$
**10**     $R = \begin{cases} 0 & \text{for terminal } s_t \\ V_\phi(s_t) & \text{for non-terminal } s_t \text{ // Bootstrap from last state} \end{cases}$
**11**     **for** $i \in \{t-1, ..., t_{start}\}$ **do**
**12**        $R \leftarrow r_i + \gamma R$
**13**        $\widehat{\log p}_{\theta,\psi}(a_i|s_i) = \alpha \hat{l}_\pi(a_i|s_i) + (1-\alpha)\hat{l}^u_\pi(a_i|s_i)$   *// Sandwich estimator*
**14**        Accumulate gradients w.r.t. $(\theta, \psi)$: $d(\theta, \psi) \leftarrow$
          $d(\theta, \psi) + (R - V_\phi(s_i))\left(\nabla_{\theta,\psi}\widehat{\log p}_{\theta,\psi}(a_i|s_i) - \frac{S_j}{l_j^2}\sum_{j\in[l]}\nabla_{\theta,\psi}\widehat{\log p}_{\theta,\psi}(a_{ij}|s_i)\right)$
**15**        Accumulate gradients w.r.t. $\phi$: $d\phi \leftarrow d\phi + \partial(R - V_\phi(s_i))^2/\partial\phi$
**16**     Perform update of $\theta$ using $d\theta$, of $\psi$ using $d\psi$ and of $\phi$ using $d\phi$
**17**     Execute ELBO updating by running Algorithm 1 with $\tau$ as input
**18**     $E \leftarrow E + 1$
**19 until** $E > E_{max}$

---

discrete space as we reveal by thorough comparison in our experiments. Another type of approach relies on being able to represent a joint probability distribution over multi-dimensional discrete actions using marginal probability distribution over each dimension (also called factored representation). Tang and Agrawal [30] apply this approach to discretized continuous control problems to decrease the learning complexity. Similarly, Delalleau et al. [7] assumes independence among dimensions to model each dimension independently. However, it is well-known in optimization and randomized allocation literature [4, 35] that dimension independence or marginal representation is not valid in the presence of constraints on the support of the probability distribution. Another type of approach [34, 36] converts the choice of multi-dimensional discrete action into a sequential choice of action across the dimensions at each time step (using a LSTM based policy network), where the choice of action in any dimension is conditioned on actions chosen for prior dimensions.

**Safe actions.** Our approach falls into the field of constrained action space in RL. Situated in the broader safe RL literature [11, 14], these works aim to constrain actions at each step for various purposes, including safety. Notably, we *do not* have an auxiliary cost that we wish to bound, thus, frameworks based on Constrained MDP [11] cannot solve our problem. Also, our state dependent constraints if written as inequalities can be extremely large and hence Lagrangian methods [2] that pull constraints into the objective are infeasible. Most of the methods that we know for constrained actions need the constraints to be written as mathematical inequalities and even then cannot handle state dependent constraints. Although action masking [17] does offer a solution for state-dependent constraints, it necessitates supplementary information in the form of masks. Thus, they do not readily apply to our problem [20]. Some of these methods[9, 24] aim at differentiating through an optimization; these methods are slow due to the presence of optimization in the forward pass and some approaches to make them faster[27] work only with linear constraints; all these approaches scale poorly in the number of inequality constraints.

**Normalizing flow policy.** The application of normalizing flows in RL has been an emerging field of study with a focus on complex policy modeling, efficient exploration, and control stability [18, 21, 33]. Normalizing flow was integrated into the Soft Actor-Critic framework to enhance the modelling expressivity beyond conventional conditional Gaussian policies and achieve more efficient exploration and higher rewards [21, 33]. In robotic manipulation, Khader et al. [18] proposed to improve the control stability by a novel normalizing-flow control structure. However, these works primarily focus

on continuous actions. To the best of our knowledge, ours is the first work that has successfully incorporated a discrete flow, extending the scope of flow policies into discrete action spaces.

## 5   Experiments

Through our experiments, we aim to address two main research questions related to our primary contributions: (1) Is the flow-based policy effective in representing categorical actions? (2) Does IAR-A2C offer advantages in constrained action space with oracle constraints? We address these two questions in Sections 5.1 and 5.2 accordingly. Following these, we present an ablation study that examines the effectiveness of various modules in our approach.

We first describe our set-up. We evaluate IAR-A2C against prior works across a diverse set of environments, including low-dimensional discrete control tasks such as CartPole and Acrobot [3], the visually challenging Pistonball task [31] with high-dimensional image inputs and an extremely large action space (upto $59,049$ categories), and an emergency resource allocation simulator in a city, referred to as Emergency Resource Allocation (ERA). CartPole and Acrobot are well-established environments. Pistonball is also a standard environment where a series of pistons on a line needs to move (up, down, or stay) to move a ball from right to left (Figure 3). While the Pistonball environment was originally designed for a multi-agent setting with each piston controlled by a distinct agent, we reconfig-

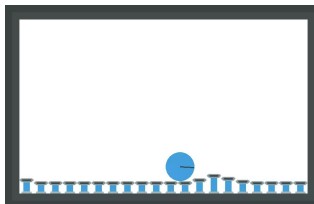

Figure 3: **Pistonball**, goal is to move the ball to the left border by operating the pistons.

ure this as a single-agent control of all pistons. This modification presents a challenging task for the central controller as the action space is exponentially large. We show results for Acrobot and three versions of Pistonball: v1 with $3^5$, v2 with $3^8$, and v3 with $3^{10}$ actions. Results for CartPole are in the appendix.

Finally, our custom environment, named ERA, simulates a city that is divided into different districts, represented by a graph where nodes denote districts and edges signify connectivity between them. An action is to allocate a limited number of resources to the nodes of the graph. A tuple including graph, current allocation, and the emergency events is the state of the underlying MDP. The allocations change every time step but an allocation action is subject to *constraints*, namely that a resource can only move to a neighboring node and resources must be located in proximity (e.g. within 2 hops on the graph) as they collaborate to perform tasks. Emergency events arise at random on the nodes, and the decision maker aims to minimize the costs associated with such events by attending to them as quickly as possible. Moreover, we explore a partially observable scenario (with no constraints) in which the optimal allocation is randomized, thus, the next node for a resource is sampled from the probability distribution over neighboring nodes that the stochastic policy represents (see appendix for set-up details). We show results for five versions of ERA: ERA-Partial with 9 actions and partial observability in unconstrained scenario, while v2 with $7^3$ actions, v3 with $8^3$ actions, v4 with $9^3$ actions, and v5 with $10^3$ actions in constrained scenario. The results for ERA-v1 are in Appendix C.

**Benchmarks**: In the unconstrained setting, we compare our approach to Wol-DDPG [10], A2C [22], factored representation (Factored, discussed in Section 4), and autoregressive approach (AR, discussed in Section 4). Wol-DDPG is chosen as it is designed to handle large discrete action spaces without relying on the dimension independence assumption. In the oracle constraints setting, we compare our method to action masking (MASK) [17], which determines the action mask by querying all actions within the action space. As far as we know action masking is currently the only existing approach for constraining action with oracle constraints, we also include a comparison with IAR-augmented AR (AR+IAR) which is able to handle the constraints by utilizing our invalid action rejection technique, as well as a comparison with Wol-DDPG to demonstrate the performance of a method that cannot enforce the constraints.

### 5.1   Learning in Categorical Action Space without Constraints

In the Acrobot environments, we use A2C as the benchmark because this task is relatively simple and A2C can easily learn the optimal policy. For the Pistonball environment, we consider both Wol-DDPG and A2C as benchmarks. The results are displayed in Figure 4. We also conduct experiments on CartPole and ERA-v5 with all constraints removed, which can be found in Appendix C.

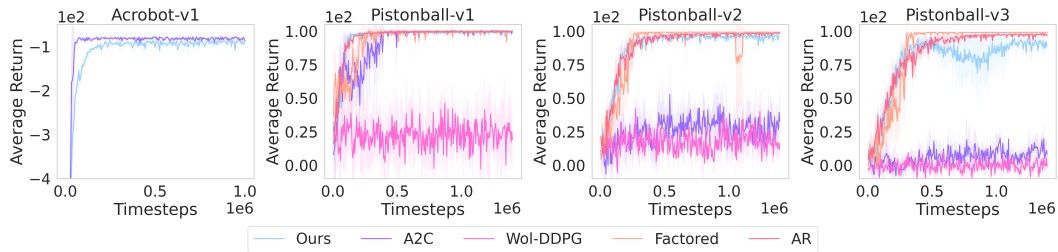

Figure 4: Learning curves over time in **(a) Acrobot** and **(b) Pistonball-v[1-3]** (without constraints). All approaches are trained with 5 seeds, except for Wol-DDPG, trained with 3 seeds due to its expensive training cost on Pistonball.

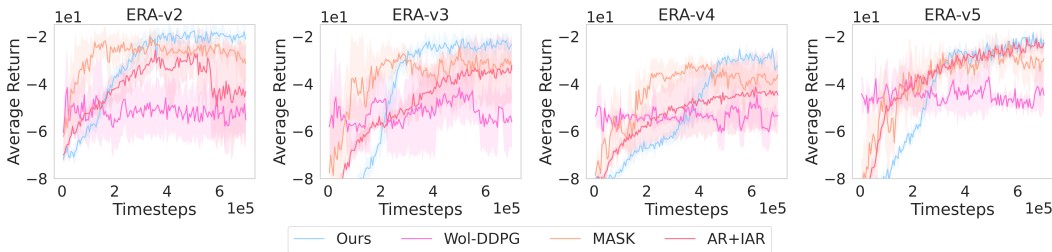

Figure 5: Learning curves in **ERA-v[2-5]** (with constraints). All settings are trained with 5 seeds.

The results of Acrobot demonstrate that the flow-based policy exhibits comparable performance to the optimal policy (A2C), which also highlights the effectiveness of our sandwich estimator for $\log \pi(a|s)$. In more challenging environments with high-dimensional image observations and extremely large action spaces, our model has comparable performance to Factored and AR, while significantly outperforming A2C and Wol-DDPG. Interestingly, we observe that Wol-DDPG struggles to learn even in the simplest Pistonball environment, despite being designed for large discrete action spaces. We hypothesize that Wol-DDPG might function properly if the actions are discrete yet ordered, as demonstrated in the quantized continuous control task presented in the Wol-DDPG paper [10].

On ERA-Partial, we demonstrate the known fact that the optimal policy in environments with partial observability may be a stochastic one [28]. We compare with the factored approach (Figure 6). In this environment, the optimal stochastic policy

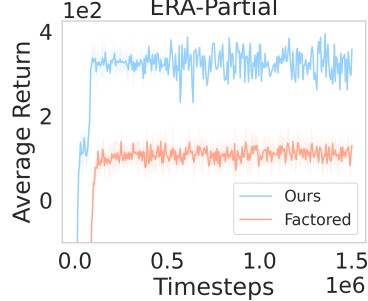

Figure 6: Learning curves in **ERA-Partial**. Our approach obtains higher return than Factored approach.

is a joint distribution that is not factorizable into a product of independent distributions over each dimension. Thus, the factored approach cannot effectively represent the joint distribution due to its independence assumption among dimensions. The experiment results show our approach significantly outperforms the factored approach. We further present a smaller example in the appendix which shows that the inability to represent an arbitrary joint distribution makes the factored approach extremely suboptimal in partial observable settings.

## 5.2 Learning in Categorical Action Space with State-Dependent Constraints

We now address the second question about constraint enforcement by IAR-A2C. The results are in Figure 5. We observe that our approach demonstrates better or comparable performance compared to the benchmarks. AR is known as a strong baseline for environments with large discrete action space, but surprisingly, it performs poorly. We hypothesize this is due to the case that the autoregressive model does not have a sense of the constraints of the remaining dimensions when it outputs the action for the first dimension, thereby producing first dimension actions that may be optimal without

constraints but are suboptimal with constraints. Detailed analysis and experimental evidence to support our hypothesis are provided in Appendix D.

Also, action masking achieves its performance by querying all actions of the entire action space, whereas our approach only requires querying a batch of actions, which is substantially smaller(e.g., $64$ versus $1,000$ for ERA-v5). Thus, while Figure 5 shows IAR-A2C taking more iteration for convergence, the same figure when drawn with the x-axis as wall clock time in Figure 7 (shown only for v4 here, others are in appendix C) shows an order of magnitude faster convergence in wall clock time. Another critical property of our approach is the guaranteed absence of constraint violations, similar to the action masking method. However, while action masking demands the full knowledge of the validity of all actions, our method merely requires the validity of the sampled actions within a batch. Note that Wol-DDPG can and does violate constraints during the learning process. Further details regarding the action violation of Wol-DDPG are provided in the appendix C.

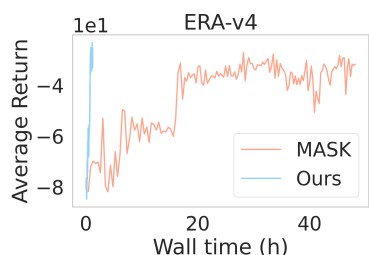

Figure 7: Learning curves for wall clock time. Our approach converges much faster than action masking.

### 5.3 Ablation Study

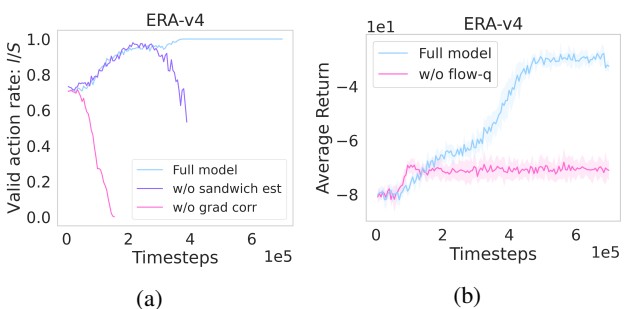

(a)                (b)

Figure 8: **(a)** Ablation of gradient correction and sandwich estimator **(b)** Ablation of posterior type.

We conduct ablation studies of IAR-A2C on ERA-v4 to investigate the effect of various choices of modules and estimators in our approach.

**Policy Gradient**: We compare the performance of approaches using the policy gradient provided by Theorem 1 (gradient correction) and the original policy gradient of A2C (standard policy gradient), while still applying invalid action rejection. We observe in Figure 8a that the number of valid actions in the batch decreases rapidly, and the program may crash if no valid actions are available.

**Sandwich Estimator**: We examine the effects if we use only the ELBO estimator for log probability of action instead of our sandwich estimator. We find that the ELBO estimator is also prone to a reduction in valid actions in Figure 8a and unstable learning as a consequence, similar to the observation when using the standard policy gradient.

**Posterior Type**: The posterior $q(z|a, s)$ can be modeled by a conditional Gaussian or a normalizing flow. In our experiments, we discover that modelling $q$ with a flow posterior is crucial for learning, as it can approximate the true posterior more effectively than a Gaussian, as seen in Figure 8b.

## 6 Conclusion and Limitations

We introduced a novel discrete normalizing flow based architecture and an action rejection approach to enforce constraints on actions in order to handle categorical action spaces with state dependent oracle constraints in RL. Our approach shows superior performance compared to available baselines, and we analyzed the importance of critical modules of our approach. A limitation of our method is in scenarios where the fraction of valid actions out of all actions is very small, and hence our sampling based rejection will need a lot of samples to be effective, making training slower. This motivates future work on improved sampling; further, better estimation of the log probability of actions is also a promising research direction for improved performance.

## Acknowledgement

This research is supported by the National Research Foundation Singapore under its AI Singapore Programme (Award Number: AISG2-RP-2020-016). Dr. Nguyen was supported by grant W911NF-20-1-0344 from the US Army Research Office.

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
