# A Proofs and Derivation

## A.1 Proof for Theorem 1

The following sequence of equations show that proof, relying on the fact that $\pi'(a|s) = \frac{\pi_\theta(a|s)}{\sum_{a_i \in \mathcal{C}(s)} \pi_\theta(a_i|s)}$. We start with the standard policy gradient for any policy $\pi'$, shown in the first line below, and then replace $\pi'(a|s) = \frac{\pi_\theta(a|s)}{\sum_{a_i \in \mathcal{C}(s)} \pi_\theta(a_i|s)}$ in the second line, followed by standard manipulation of the log function.

$$\nabla_\theta J(\theta) = \mathbb{E}_{\pi'}\left[Q^{\pi'}(s,a)\nabla_\theta \log \pi'(a|s)\right] \tag{8}$$

$$= \mathbb{E}_{\pi'}\left[Q^{\pi'}(s,a)\nabla_\theta \log \frac{\pi_\theta(a|s)}{\sum_{a_i \in \mathcal{C}(s)} \pi_\theta(a_i|s)}\right] \tag{9}$$

$$= \mathbb{E}_{\pi'}\left[Q^{\pi'}(s,a)\nabla_\theta \log \pi_\theta(a|s) - Q^{\pi'}(s,a)\nabla_\theta \log \sum_{a_i \in \mathcal{C}(s)} \pi_\theta(a_i|s)\right] \tag{10}$$

$$= \mathbb{E}_{\pi'}\left[Q^{\pi'}(s,a)\nabla_\theta \log \pi_\theta(a|s) - Q^{\pi'}(s,a)\frac{\sum_{a_i \in \mathcal{C}(s)} \nabla_\theta \pi_\theta(a_i|s)}{\sum_{a_i \in \mathcal{C}(s)} \pi_\theta(a_i|s)}\right] \tag{11}$$

## A.2 Proof for Lemma 1

Fix state $s$ and consider a function $F(a) = \begin{cases} \frac{\nabla_\theta \pi_\theta(a|s)}{\pi_\theta(a|s)} & \text{for } a \in \mathcal{C}(s) \\ 0 & \text{otherwise} \end{cases}$. Then,

$$\mathbb{E}_\pi[F(a)] = \sum_{a \in \mathcal{C}(s)} \nabla_\theta \pi_\theta(a|s)$$

Thus, if we obtain a sample average estimate for $\mathbb{E}_\pi[F(a)]$ then it is an unbiased estimate for $\sum_{a \in \mathcal{C}(s)} \nabla_\theta \pi_\theta(a|s)$. For $S$ samples from $\pi$ with $l$ being valid samples, the sample average estimate for $\mathbb{E}_\pi[F(a)]$ is $\frac{1}{l}\sum_{j \in [l]} \nabla_\theta \log \pi_\theta(a_j|s)$.

Similarly, for the next estimate, consider a function $G(a) = \begin{cases} 1 & \text{for } a \in \mathcal{C}(s) \\ 0 & \text{otherwise} \end{cases}$. Clearly, then $\mathbb{E}_\pi[G(a)] = \sum_{a \in \mathcal{C}(s)} \pi_\theta(a|s)$ and a sample avergae estimate of $\mathbb{E}_\pi[G(a)]$ is $\frac{l}{S}$.

## A.3 Soft Threshold Function

In Argmax Flow [16], a threshold function was introduced to enforce the argmax constraints, i.e. the variational distribution $q(v|x)$ should have support limited to $\mathcal{S}(x) = \{v \in \mathbb{R}^{D \times K} : x = \arg\max v\}$. The thresholding-based $q(v|x)$ was defined by Alg. 3 in Argmax Flow. However, the formula to evaluate $\det dv/du$ is not given, which is essential when estimating ELBO $\hat{l}_\pi$ and CUBO $\hat{l}_\pi^u$. We derive it here.

Let's follow the notations in Alg. 3 of Argmax Flow. Suppose index $i$ is the one that we want to be largest ($i$ is a fixed index). The soft threshold function is given by

$$v_j = u_i - \log(1 + e^{u_i - u_j})$$

Note that the threshold $T = v_x = u_x$ (we cannot use $v_x$ to define $v$ itself, so $T$ is $u_x$). Then,

- if $j = i$ then $v_j = u_i, \frac{\partial v_j}{\partial u_k} = 1$
- if $k \neq j$ or $k \neq i$ then $\frac{\partial v_j}{\partial u_k} = 0$
- if $k = i$ then $\frac{\partial v_j}{\partial u_k} = 1 - \frac{1}{1+e^{u_i - u_j}} \times e^{u_i - u_j}$
- if $k = j$ then $\frac{\partial v_j}{\partial u_k} = \frac{1}{1+e^{u_i - u_j}} \times e^{u_i - u_j}$

Table A.1: Various Types of Alpha $\alpha$

| Types of $\alpha$ | Remark |
|---|---|
| Static $\alpha$ | $\alpha$ is fixed to be 0.5 |
| Trainable $\alpha$ | $\alpha$ is a trainable parameter, updated by the policy gradient |
| Adaptive $\alpha(\hat{l}_\pi, \hat{l}_\pi^u)$ | $\alpha$ is conditioned on the ELBO and the CUBO |

$\det \mathrm{d}\boldsymbol{v}/\mathrm{d}\boldsymbol{u}$ is a $K \times K$ determinant, where only the elements on the diagonal and on the column $i$ is non-zero, other elements are zero. We can unfold the determinant by the i-th row. Finally, we have

$$\det \mathrm{d}\boldsymbol{v}/\mathrm{d}\boldsymbol{u} = \prod_{j=1, j \neq i}^{K} \frac{\partial v_j}{\partial u_j} = \prod_{j=1, j \neq i}^{K} \mathrm{sigmoid}(u_i - u_j)$$

# B   Experimental Details

## B.1   Effect of the Sandwich Estimator's Weight $\alpha$

The value of $\alpha$ plays a crucial role in estimating the log probability, which in turn impacts the performance of the model. Various types of $\alpha$ are presented in Table A.1. Through our experiments, we have observed that an adaptive $\alpha$ yields greater stability and, in certain environments, leads to improved performance. This is illustrated in Figure A.1, where the adaptive $\alpha$ exhibits superior performance compared to the static value of $\alpha = 0.5$.

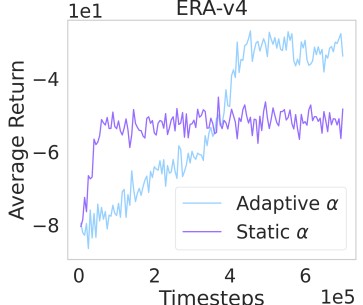

Figure A.1: Performance with adaptive $\alpha$ and static $\alpha$ in **ERA-v4**.

## B.2   Effect of Invertible Functions $F_w$

We have explored different types of invertible function $F_w$ (called latent flow model) in our study, including affine coupling bijections [2], as well as more advanced models such as AR Flow and Coupling Flow, as described in section B.1 of Argmax Flow [16]. The AR Flow and Coupling Flow methods offer enhanced capabilities for modeling complex distributions, and they have been successfully applied to language modeling tasks within the Argmax Flow framework. However, through our experiments, we have observed that even a simple latent flow model is sufficient for achieving good performance and exhibits faster convergence. We attribute this finding to the fact that the increased parameterization in AR Flow requires a larger amount of training data to effectively learn.

Table A.2: Optimization details

| Environment | n_envs | lr | Optimizer | batch size |
|---|---|---|---|---|
| CartPole | 8 | 3e-4 | RMSprop | 256 |
| Acrobot | 8 | 3e-4 | RMSprop | 256 |
| ERA-v5 w/o cstr | 64 | 3e-4 | RMSprop | 512 |
| Pistonball-v[1-3] w/o cstr | 64 | 3e-4 | RMSprop | 512 |
| ERA-v[1-5] w/ cstr | 64 | 3e-4 | RMSprop | 256 |
| Pistonball-v[1-2] w/ cstr | 64 | 3e-4 | RMSprop | 512 |

## B.3   Optimization Details

The total timesteps of training for each environment are determined based on the convergence of our model and benchmarks. Typically, we train each setting using 5 different random seeds, unless

---

[2]We exploit the implementation from: https://github.com/didriknielsen/survae_flows

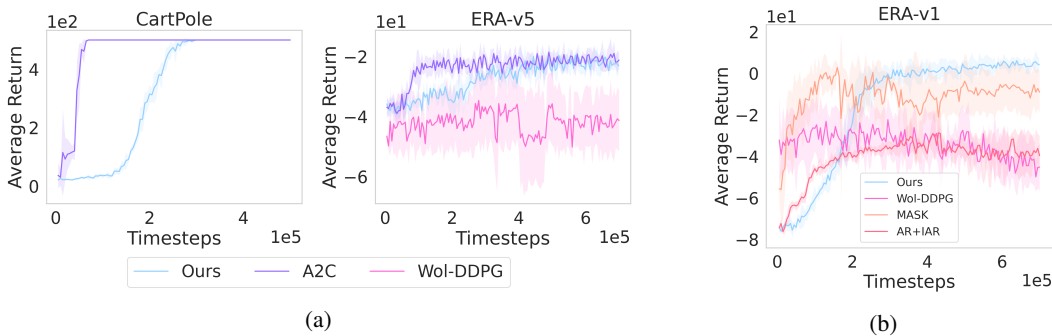

(a)                                                    (b)

Figure A.2: **(a)** Learning curves in **CartPole** and **ERA-v5** (without constraints); **(b)** Learning curves in **ERA-v1** (with constraints).

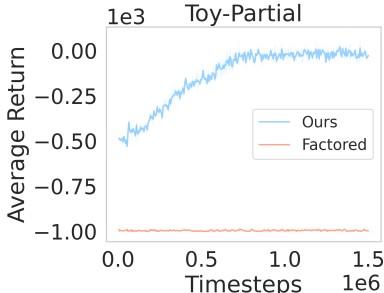

Figure A.3: Learning curves in **Toy-Partial**.

otherwise specified. When evaluating the model's performance at a specific timestep with a specific seed, we employ a separate set of 10 testing environments and report the mean return over these environments. Further details can be found in Tables A.2. Note that `n_envs` denotes the number of environments running in parallel, `lr` denotes the learning rate, and `batch size` refers to the batch size when we execute ELBO updating (Algorithm 1 in the main paper). Furthermore, we will make the code used to reproduce these results publicly available.

### B.4    Range of Considered hyperparameters

We conducted experiments varying the number of samples used for estimating $\log \pi(a|s)$, specifically considering the values $\{1, 2, 4, 8\}$, as well as the inclusion of reward normalization. We find that using 2 or 4 samples generally leads to good performance across most of our experiments.

### B.5    Network structure

Our implementation is built on Stable-Baseline3 [25]. In different environments, different state encoders were exploited. We used MLP encoder for discrete control tasks and CNN encoder for Pistonball task. In ERA environment, a customized state encoder was applied to handle the graph state based on the implementation from [19].

### B.6    Computational resources

Experiments were run on NVIDIA Quadro RTX 6000 GPUs, CUDA 11.0 with Python version 3.8.13 in Pytorch 1.11.

## C    Additional Experiment

In this section, we present additional experimental results obtained from our study.

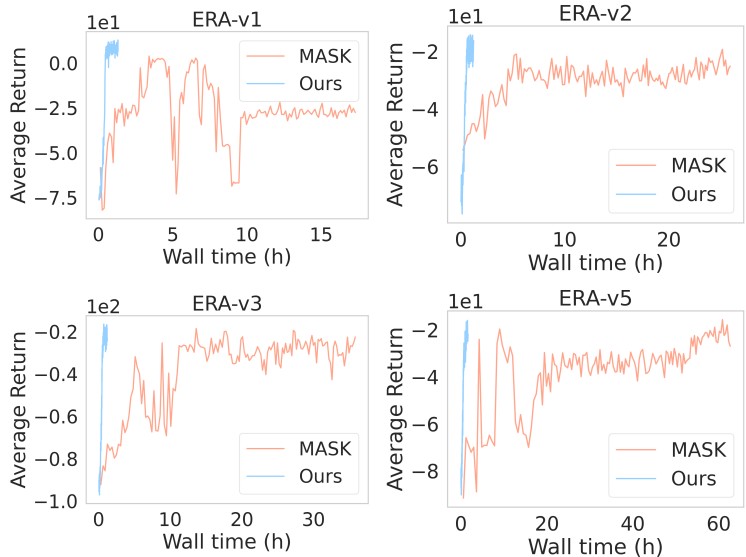

Figure A.4: Learning curves for wall clock time.

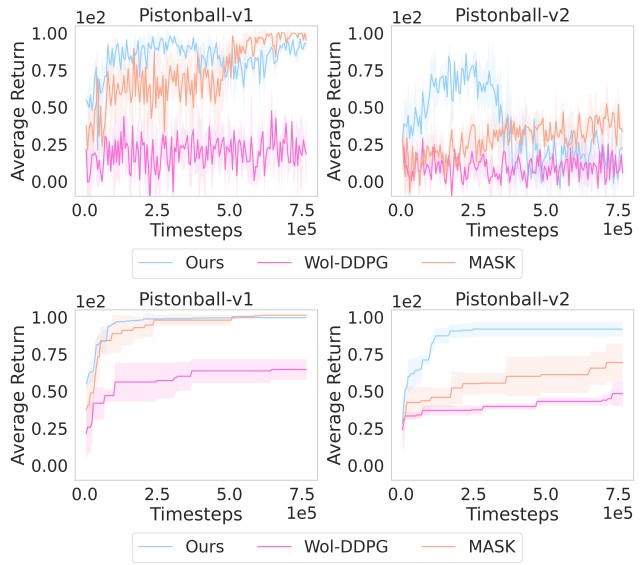

Figure A.5: Top: Learning curves in **Pistonball-v1** and **v2** (with constraints); Bottom: Best-till-now returns in **Pistonball-v1** and **v2**.

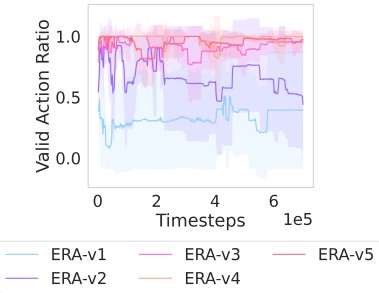

Figure A.6: Valid action ratio of Wol-DDPG during training.

In the **non-constrained setting**, we conduct experiments on CartPole, ERA-v5, and a toy environment with partial observability - Toy-Partial. For ERA-v5, we remove all constraints, allowing for the allocation of resources in any nodes. Figure A.2a illustrates that our approach achieves comparable performance to the best benchmark when the action space is not excessively large. Toy-Partial is built exactly on the example described in Section 3.1 of [28], in which the optimal stochastic policy can be arbitrarily better than any deterministic policy. We only modify the setting by augmenting the dimensionality of actions. There are two actions A and B in their example, while we use a 2-dimensional representation ((0, 1) representing action A, (1, 0) representing action B, (1, 1) and (0, 1) staying at the current state.) to simulate the case in multi-dimensional action space. Figure A.3 shows that our approach can perform significantly better than Factored approach in Toy-Partial.

In the **constrained setting**, we perform additional experiments on ERA-v1, Pistonball-v1, and Pistonball-v2. In the Pistonball environment, we introduce a constraint that restricts the upward movement of pistons on the left side to ensure the ball continues rolling to the left. Our experiments demonstrate that this constraint presents significant difficulty when the action space is large.

Regarding ERA-v1, our model outperforms the benchmark, as depicted in Figure A.2b. Additionally, we analyze the average return over wall clock time for ERA-v1, v2, v3, and v5, and our model exhibits an order of magnitude faster convergence, as illustrated in Figure A.4.

For Pistonball, our model demonstrates comparable performance in the smaller environment (Pistonball-v1) and superior performance in the larger environment (Pistonball-v2) where the benchmarks struggle to effectively learn. These findings are presented in Figure A.5, which includes the average return over timesteps in the top plot and the best-till-now returns in the bottom plot. By best-till-now we mean the best evaluated return till the current timestamp, which is a commonly used metric particularly when the return done not increase monotonically over time steps and hence the best model might be an intermediate one.

We also observed that our approach can learn a stochastic optimal policy in ERA environment, which corresponds to our motivation that a stochastic policy is preferred in many resource allocation problems. For example, we have observed that in ERA-v4, a stochastic optimal allocation was learned by our approach in a given state (shown as a distribution of 200 sampled actions (Table A.3).

Finally, we investigate the **constraint violation of Wol-DDPG** across ERA-v1 to v5 in Figure A.6. Our observations reveal that the constraint violation of Wol-DDPG can be significant, with a valid action ratio reaching only 15% in the worst-case scenario. Note that the valid action ratio mentioned here specifically pertains to the ratio of valid actions generated by the agent (the action output by the policy at each timestep) to all the actions output in a single episode, which differs from the valid action rate discussed in the ablation study in the main paper. In the ablation study shown in Figure 8a, the valid action rate refers to the fraction ($l/S$) of valid actions $l$ within the $S$ samples at a particular timestep in IAR framework shown in Figure 1. Thus, valid action rate is metric specific to our framework; note that valid action ratio for our framework is always 1 as IAR always outputs valid actions.

## D AR in Constrained Scenario

We have observed that AR approach does not perform well in the constrained scenario. We give our analysis and experimental evidence here.

First, we describe our specific ERA set-up. We are aiming to allocate 3 resources to 9 areas with the 9 areas lying on a graph (we do not show the graph here). The constraints are that in any allocation, R2 and R3 must be within 2 hops (inclusive) of each other.

Each allocation of resource R1, R2, R3 is given as a vector, e.g., (4, 0, 3) is the allocation of R1 to area 4, R2 to area 0, R3 to area 3. In a AR approach there is a dimension dependency in the policy network. Here allocation of R2 (output by a neural network, which we call R2 network) depends on allocation of R1 and allocation of R3 (output of R3 network) depends on allocation of R2 and R1. The R1 network, which outputs location of R1, is conditionally independent (note that R1, R2, R3 networks use the shared weights).

Shown below is an optimal allocation learned by **our approach** in state 1 (current allocation (8, 2, 8)), shown as a distribution of 200 sampled actions (Table A.3).

Table A.3: Optimal policy (ours) in state 1. Showing actions with top-3 probability.

| Action | Number | Distribution |
|---|---|---|
| (4, 0, 3) | 100 | 0.5 |
| (4, 0, 6) | 100 | 0.5 |
| (0, 0, 0) | 0 | 0.0 |
| ... | ... | ... |

Shown below is the best allocation learned by the **AR approach** in state 1, shown as a distribution of 200 sampled actions (Table A.4).

Table A.4: AR policy in state 1.

| Action | Number | Distribution |
|---|---|---|
| (3, 6, 6) | 173 | 0.865 |
| (3, 3, 6) | 17 | 0.085 |
| (6, 3, 6) | 5 | 0.025 |
| ... | ... | ... |

However, the above is not optimal in state 1. But, if we fix R1 to be in area 4 and R2 to be in area 0 and provide that as forced input to the R3 network, then we get the Table A.5 from the AR network.

Table A.5: AR policy in state 1 after fixing R1 in area 4 and R2 in area 0.

| Action | Number | Distribution |
|---|---|---|
| (4, 0, 6) | 164 | 0.82 |
| (4, 0, 3) | 36 | 0.18 |
| (0, 0, 0) | 0 | 0.00 |
| ... | ... | ... |

This shows that R3 network has learned a good policy, but the R1 network is unable to independently produce the area 4 that could trigger the optimal output. We conjecture this could be due to fact that R1 is not aware of constraints since the constraint can be solely enforced by R2 network and R3 network. Hence, R1 might be outputting areas that can be optimal if there were no constraints. We have further evidence of the same happening in another state:

Shown below is an optimal allocation learned by **our approach** in state 2 (current allocation (1, 7, 5)), shown as a distribution of 200 sampled actions (Table A.6).

Table A.6: Optimal policy (ours) in state 2.

| Action | Number | Distribution |
|---|---|---|
| (4, 0, 3) | 200 | 1.0 |
| (4, 0, 6) | 0 | 0.0 |
| (0, 0, 0) | 0 | 0.0 |
| ... | ... | ... |

Shown in Table A.7 is the best allocation learned by the **AR approach** in state 2, shown as a distribution of 200 sampled actions. Again, if we fix R1 to be in area 4 and R2 to be in area 0 and provide that as forced input to the R3 network, then we get the Table A.8 from the AR network. This supports our claim that the R1 network finds it difficult to reason about constraints.

While this is not a complex example and may be resolved through another ordering, we wish to highlight that constraints can also be complex and they will always introduce issues with ordering dependent approaches. To handle such issues, one may need knowledge on the dimension dependency, or make efforts on trying different orders of generating the allocations.

Table A.7: AR policy in state 2.

| Action | Number | Distribution |
|--------|--------|--------------|
| (3, 6, 3) | 81 | 0.405 |
| (3, 6, 6) | 81 | 0.405 |
| (6, 6, 3) | 16 | 0.008 |
| ... | ... | ... |

Table A.8: AR policy in state 2.

| Action | Number | Distribution |
|--------|--------|--------------|
| (4, 0, 3) | 197 | 0.985 |
| (4, 0, 1) | 2 | 0.010 |
| (4, 0, 2) | 1 | 0.005 |
| ... | ... | ... |

# E   Details of Emergency Resource Allocation Environment

We describe the details of our custom environment ERA in this section. This environment simulates a city that is divided into different districts, represented by a graph where nodes denote districts and edges signify connectivity between them. An action is to allocate a limited number of resources to the nodes of the graph. A tuple including graph, current allocation, and the emergency events is the state of the underlying MDP. The allocations change every time step but an allocation action is subject to constraints, namely that a resource can only move to a neighboring node and resources must be located in proximity (e.g. within 2 hops on the graph) as they collaborate to perform tasks. Emergency events arise at random on the nodes and the decision maker aims to minimize the costs associated with such events by attending to them as quickly as possible. Finally, the optimal allocation is randomized, thus, the next node for a resource is sampled from the probability distribution over neighboring nodes that the stochastic policy represents.

Each version of the ERA environment is characterized by an adjacency matrix that defines the connectivity of districts within the simulated city and a cost matrix that quantifies the expenses associated with traversing from one node to another. The agent's performance is evaluated based on the successful resolution of emergency events, leading to rewards, while penalties are incurred for failure to address the emergency. The agent's utility at each timestep encompasses the reward (or penalty) and the negative moving costs. To increase the complexity of the task, we introduce different types of emergency events. Each event type follows a fixed distribution over the nodes, and the event type itself is determined by a categorical distribution. For example, a distribution of [0.3, 0.7] means that 30% of the events are of type 1 and 70% are of type 2. The specifics of each version are presented in Table A.9, where the columns # rsc, # nodes, and hops represent the number of resources, the number of nodes in the graph, and the maximal allocation distance between two resources, respectively.

ERA-Partial is one setting with partial observability, which has three states but one possible observation. The RL agent is encouraged to change its (unobserved) state by obtaining a reward, otherwise it obtains a penalty. To perform well, the RL agent needs to perform stochastically. The ERA implementation, including configuration file that consists of the adjacency matrix, cost matrix and other relevant parameters, will be made available for reproducibility purposes.

Table A.9: Specifics for ERA

| Version | # rsc | # nodes | hops |
|---------|-------|---------|------|
| partial | 2 | 3 | 1 |
| v1 | 3 | 6 | 1 |
| v2 | 3 | 7 | 1 |
| v3 | 3 | 8 | 2 |
| v4 | 3 | 9 | 2 |
| v5 | 3 | 10 | 3 |