# OpenReview forum: "Generative Modelling of Stochastic Actions with Arbitrary Constraints in Reinforcement Learning"
_NeurIPS.cc/2023/Conference — NeurIPS 2023 poster_

### Official Review · Reviewer_Xx2Q · 2023-07-04

**Soundness:** 2 fair
**Presentation:** 3 good
**Contribution:** 3 good
**Rating:** 6
**Confidence:** 4

**Summary:**

This paper introduces an algorithm for learning with large discrete action spaces where only some actions are valid for any given state in ways that are hard to programmatically exclude. The paper proposes to use normalizing flows to learn a policy across such action spaces without having to model the full action probability distribution, which is difficult to estimate and slow to learn. Algorithmic details for how to implement such a flow policy are discussed, such as the use of a sandwich bound for better action log probability estimation, and an oracle action validator to constrain invalid actions and reweight valid ones.

**Strengths:**

This paper sets out the goal of improving RL training efficiency in the context of very large discrete action spaces without assuming any structure or ordering among the actions (though it optionally assumes an action validity checker when some actions may be invalid for some states). Very large action spaces are a known issue impeding learning, typically solved by assuming domain-specific structure or allowing  e.g. step-by-step action composition, which may be unsuitable sometimes and are inelegant in a sense, so this problem is worth addressing.

The paper itself is largely well written. I generally was able to follow the arguments and math as presented, with a few notable exceptions discussed below.

The approach presented for tackling this problem seems intuitive and draws on established algorithms like normalizing flows and ELBO/CUBO in a nontrivial way. The authors argue for the use of normalizing flows among other generative models here, and their argument seems persuasive as a nonexpert in generative modelling.



**Weaknesses:**

First and foremost, the current abstract needs to change. It does a very poor job summarizing the content of the paper, and seems bizarrely out of place compared to the rest of the paper, which is mostly well written and clear. I put this in "weaknesses" rather than "questions" because this seems like a major issue that needs to be resolved prior to publication.

Related tangentially to the abstract, the authors assert several times that the optimal policy needs to be stochastic for the domain they are interested in. I'm unsure why this is asserted, since it's not obvious that that is the case for the custom evaluation domain. Perhaps there's something here that I'm not seeing, but this seems like an out of place insistence that's not relevant to the core problem of very large discrete action spaces. It's useful to have a stochastic policy during training, but I'm not sure offhand what conditions would lead to the optimal policy being stochastic, especially in the discrete action domain.

Segueing nicely into another concern I have, that assertion (the optimal policy must be stochastic) is used to dismiss one approach that is commonly used to address very large action spaces, to decompose a single action decision into a sequence of decisions with each decision conditioned on the previous decisions for that action. I'm sympathetic to the idea that this isn't a perfect solution (it's obviously nicer to act directly in the primitive action space), but this doesn't seem like grounds to dismiss such an approach and not benchmark against it.

The last of my major concerns is that the main comparison to prior work, versus Wol-DDPG, is not valid because Wol-DDPG assumes spatial structure among discrete actions, which the authors briefly mention as an issue in section 5.1 and then ignore. It's unclear whether Wol-DDPG was used correctly here, particularly since all the learning curves shown for that algorithm are flat- the agent does not improve over the course of training on any variant of either task. Absent that comparison, the experiments clearly show that IAR-A2C outperforms vanilla A2C and is much less computationally expensive than action masking on the two tasks respectively, but I'm concerned by the lack of comparison to previous work given that this isn't a new problem.

I'll also register that the sample-and-reject approach to action validity used is somewhat unsatisfying and a "weakness", but this is acknowledged by the authors as a limitation of the current algorithm and I'm unsure what a better non-domain-specific solution would be, so I'm not holding that against this paper as it is.

Given all the above, while I like the core concept here, I'm hesitant to recommend acceptance as is. The algorithm seems sound, but as the problem of large action spaces is chiefly a practical (rather than theoretical) one I feel a solid performance comparison to prior methods (or a more convincing argument for why such a comparison is invalid) is needed to validate the algorithm.



**Questions:**

Some other suggestions (in rough decreasing order of importance):

-I get why without the gradient correction the policy gets stuck in section 5.3, but it's not obvious to me why this happens with ELBO versus sandwich bounds as well since ELBO should be biased but consistent similar to sandwich (which should be a better bound, but theoretically ELBO should still converge, no?). This result should be explained a little more (and I'd also appreciate ELBO versus sandwich on e.g. pistonball to show the difference when valid action issues don't apply).

-I got a little lost in the details and notation at the end of section 3.1. While I get the gist of things, this explanation of the structure of a flow policy and figure 2 illustrating it seem somewhat confusing and could be improved, since this is one of the core algorithmic contributions of the paper.

-Theorem 1 seems like it could be stated more simply. This is just reweighting the gradients based on the fraction of sampled actions which are valid, no? I'd argue that doesn't need a whole theorem to justify (unless I'm missing something here)

-Please move the symbol definitions from the last paragraph of section 3.2 to the algorithm 2 box, it's awkward having these definitions stated in the text rather than in the box. Also adding other givens/definitions to the box would be nice as well.

-There's a couple places where several things are listed followed by "etc." I'd suggest removing the "etc" since it looks too casual stylistically.

-There's several spots of awkward grammar/phrasing throughout the paper. Not enough to impede understanding, but I'd recommend an editing pass to clean them up before camera-ready.

**Limitations:**

In general the limitations of the presented algorithm are clearly stated (questions about comparison to prior work above aside). There's clearly room to improve with valid action handling, but this is stated by the authors, and it's a complex enough problem that leaving it to future work is valid.

Regarding negative social impact, I don't think the core algorithm has any issues. The specific application example of routing emergency services between hotspots and potentially triaging between different incidents does have non-trivial moral hazard, and using the system to coordinate police response to (for example) protests has high risk for negative societal impacts. However, as that application seems to be mostly illustrative and it's not clear that the authors intend their algorithm to be used directly in the real world I think the paper itself is fine as it is (though I might suggest a different example use case for future work to avoid such concerns if possible).

---

> ### Author Rebuttal · Authors · 2023-08-10
>
> We thank the reviewer for the comments and feedback.
>
> We will write an alternate abstract and present it to the reviewer in the discussion phase (we are just not sure if that is allowed in this first round of rebuttal).
>
> In allocation problems, stochastic allocation, if one exists, is often preferred over deterministic ones due to reasons of fair tie breaking and not starving a recipient of resources (please see Introduction in [Budish et al. 2013]). In a sequential decision making problem, a stochastic optimal policy can exist in a MDP if there are two or more actions that maximize the Q value in a state. Further, with partial observability there can be scenarios where the optimal policy is stochastic and this optimal stochastic policy is better than any deterministic policy (see Fact 2 in [Singh et al. 1994]).
>
> [Budish et al. 2013] Eric Budish, Yeon-Koo Che, Fuhito Kojima, and Paul Milgrom. Designing random allocation mechanisms: Theory and applications. American economic review, 103(2):585–623, 2013.
>
> [Singh et al. 1994] Singh, Satinder P., Tommi Jaakkola, and Michael I. Jordan. "Learning without state-estimation in partially observable Markovian decision processes." In Machine Learning Proceedings 1994, pp. 284-292. Morgan Kaufmann, 1994.
>
> We have implemented an autoregressive model (unfortunately, we did not find any publicly available code) and results are in the attached pdf; also present are results for the marginal/factored approach.
>
> Figure 1 in the attached PDF shows that our approach has similar performance as the factored/marginal and the autoregressive approach in the **unconstrained case**, which is as per expectation since the dimensions are independent and the factored approach assumes underlying independence across dimensions. The autoregressive approach assumes a particular dependence structure (based on ordering of dimensions) but is equivalent to marginal A2C when there is underlying independence across dimensions.
> Figure 3 in the PDF shows that the autoregressive approach does not perform well in the **constrained case**. For this experiment, we used our IAR approach to enforce constraints, as the autoregressive approach by itself does not enforce constraints. Note that IAR is a general method that can be applied to a wide range of unconstrained RL algorithms.
>
> We have additional experiment still running and hope to share these during the discussion phase
>
> Response to questions:
>
> ELBO vs sandwich: Yes, indeed ELBO is consistent but we never acquire as many samples in one step (note that there is a different ELBO every state) for ELBO to reach close to the true log probability at that time step. Moreover, during training the true log probability is also changing as we update the policy using the policy gradient. Thus, unlike a standard generative modeling set-up where the ground truth distribution is fixed, we are chasing a changing ground truth - hence, any better estimate (with limited scope of samples) should intuitively provide an advantage. We have added results for the use of only ELBO in unconstrained case to the attached pdf; Figure 2 in the attached pdf shows that only ELBO still has instability issues even in the unconstrained case.
>
> We will improve the discussion as suggested by the reviewer.

---

> > ### Comment · Reviewer_Xx2Q · 2023-08-11
> > **Response to rebuttal**
> >
> > Thanks for the clarifications and references on stochastic optimal policies. Do these conditions necessarily apply to the ERA tasks? I'm still a little concerned that this consideration is not core to the problem being addressed and could be removed or reduced in the paper.
> >
> > The results for the autoregressive baseline are interesting- do you have any intuition for why it performed worse in the constrained case? Is this simply due to the lack of dependence among action dimensions? Given the presence of constraints, I'd expect there is still some, structure, no? What about constrained actions makes the autoregressive model worse given you use IAR for both methods?
> >
> > That result is encouraging, though I'd like to see it fleshed out/explained a little more if possible before I improve my score (short timelines make that tricky, I know).

---

> > > ### Author Response · Authors · 2023-08-12
> > >
> > > Thank you for your interesting comment. We spent some time investigating the AR approach and we present our analysis (supported with some experimental evidence below). First, we describe our specific ERA set-up. We are aiming to allocate 3 resources to 9 areas with the 9 areas lying on a graph (we do not show the graph here). The constraints are that in any allocation, R2 and R3 must be within 2 hops (inclusive) of each other.
> > >
> > > Each allocation of resource R1, R2, R3 is given as a vector, e.g., (4, 0, 3) is the allocation of R1 to area 4, R2 to area 0, R3 to area 3. In a AR approach there is a dimension dependency in the policy network. Here allocation of R2 (output by a neural network, which we call R2 network) depends on allocation of R1 and allocation of R3 (output of R3 network) depends on allocation of R2 and R1. The R1 network, which outputs location of R1, is conditionally independent (note that R1, R2, R3 networks use the shared weights).
> > >
> > > Shown below is an optimal allocation learned by **our approach** in state 1 (current allocation (8, 2, 8)), shown as a distribution of 200 sampled actions (Table 1).
> > >
> > > Table 1. Optimal policy (ours) in state 1. Only showing actions with the highest probability (Top 3).
> > >  **Action** | **Number** | **Distribution**
> > > :---:|:---:|:---:
> > >  (4, 0, 3)  | 100 | 0.5
> > >  (4, 0, 6)  | 100 | 0.5
> > >  (0, 0, 0)  | 0 | 0.0
> > >  ... | ... | ...
> > >
> > > (Note the optimal policy learned is stochastic)
> > >
> > > Shown below is the best allocation learned by the **AR approach** in state 1, shown as a distribution of 200 sampled actions (Table 2).
> > >
> > > Table 2. AR policy in state 1.
> > >  **Action** | **Number** | **Distribution**
> > > :---:|:---:|:---:
> > >  (3, 6, 6)  | 173 | 0.865
> > >  (3, 3, 6)  | 17 | 0.085
> > >  (6, 3, 6)  | 5 | 0.025
> > >  ... | ... | ...
> > >
> > > However, the above is not optimal in state 1. But, if we fix R1 to be in area 4 and R2 to be in area 0 and provide that as forced input to the R3 network, then we get the following from the AR network:
> > >
> > > Table 3. AR policy in state 1 after fixing R1 in area 4 and R2 in area 0.
> > >  **Action** | **Number** | **Distribution**
> > > :---:|:---:|:---:
> > >  (4, 0, 6)  | 164 | 0.82
> > >  (4, 0, 3)  | 36 | 0.18
> > >  (0, 0, 0)  | 0 | 0.00
> > >  ... | ... | ...
> > >
> > > This shows that R3 network has learned a good policy, but the R1 network is unable to independently produce the area 4 that could trigger the optimal output. We conjecture this could be due to fact that R1 is not aware of constraints since the constraint can be solely enforced by R2 network and R3 network. Hence, R1 might be outputting areas that can be optimal if there were no constraints. We have further evidence of the same happening in another state:
> > >
> > > Shown below is an optimal allocation learned by **our approach** in state 2 (current allocation (1, 7, 5)), shown as a distribution of 200 sampled actions (Table 4).
> > >
> > > Table 4. Optimal policy (ours) in state 2.
> > >  **Action** | **Number** | **Distribution**
> > > :---:|:---:|:---:
> > >  (4, 0, 3)  | 200 | 1.0
> > >  (4, 0, 6)  | 0 | 0.0
> > >  (0, 0, 0)  | 0 | 0.0
> > >  ... | ... | ...
> > >
> > > Shown below is the best allocation learned by the **AR approach** in state 2, shown as a distribution of 200 sampled actions (Table 5).
> > >
> > > Table 5. AR policy in state 2.
> > >  **Action** | **Number** | **Distribution**
> > > :---:|:---:|:---:
> > >  (3, 6, 3)  | 81 | 0.405
> > >  (3, 6, 6)  | 81 | 0.405
> > >  (6, 6, 3)  | 16 | 0.008
> > >  ... | ... | ...
> > >
> > > Again, if we fix R1 to be in area 4 and R2 to be in area 0 and provide that as forced input to the R3 network, then we get the following from the AR network:
> > >
> > > Table 6. AR policy in state 2 after fixing R1 in area 4 and R2 in area 0.
> > >  **Action** | **Number** | **Distribution**
> > > :---:|:---:|:---:
> > >  (4, 0, 3)  | 197 | 0.985
> > >  (4, 0, 1)  | 2 | 0.010
> > >  (4, 0, 2)  | 1 | 0.005
> > >  ... | ... | ...
> > >
> > > This supports our claim that the R1 network finds it difficult to reason about constraints.
> > >
> > > While this is not a complex example and may be resolved through another ordering, we wish to highlight that constraints can also be complex and they will always introduce issues with ordering dependent approaches. To handle such issues, one may need knowledge on the dimension dependency, or make efforts on trying different orders of generating the allocations.
> > >
> > > Finally, we note that Table 1 above shows that our approach learns stochastic policies, if one exists, which does occur in ERA problems. While not core to the problem being addressed, a stochastic policy is preferred in allocation problems; even in allocating emergency resources we may not wish to allocate ambulances to one area always (in some state) and hence depriving another equally deserving area (in this state). Thus, we consider it a feature that we output stochastic policies.

---

> > > > ### Comment · Reviewer_Xx2Q · 2023-08-14
> > > > **Additional Suggestions**
> > > >
> > > > Interesting- If I understand the above right that would seem to suggest the action constraints are the source of the autoregressive model's underperformance, but you specified the AR model used your IAR algorithm to handle constraints, so shouldn't constraints be handled identically to IAR-A2C and thus not a source of performance loss?
> > > >
> > > > This discussion of constraint satisfaction seems important, and I'd encourage you to work up a concise version that can go in the paper to highlight and explain the performance gap between AR and IAR-A2C for the next revision. It sounds like the constrained case is the main problem this paper solves (given AR performs comparably on the unconstrained case), so spending more space exploring why and how that works seems warranted.

---

> > > > > ### Author Response · Authors · 2023-08-15
> > > > >
> > > > > Thank you for your additional suggestions. Yes, IAR ensures constraints are never violated for both auto-regressive (AR) and our approach. But, constraint satisfaction seems to have a detrimental effect on reward maximization for AR based on what we saw in the example above. In AR there is a particular dependency structure assumed: $P(a_1,a_2,a_3) = P(a_1)P(a_2|a_1)P(a_3|a_2,a_1)$ with each of $P(a_1)$, $P(a_2|a_1)$, and $P(a_3|a_2,a_1)$ represented by a neural network respectively ($a_i$ denotes the allocation of the resource $R_i$). Intuitively, the constraint satisfaction for the type of constraint (across dimensions) that we are using is mostly (and can be) ensured by the neural networks that represent the dependencies $P(a_2|a_1)$ and $P(a_3|a_2,a_1)$, which implies that the neural network representing $P(a_1)$ is constraint unaware. What this results in is that the neural network outputting $a_1$ does so without regard to constraints. Then, if $(a_1^*, a_2^*, a_3^*)$ is the optimal action under constraints, the neural network for $P(a_1)$ might output $a_1'$ as this network considers the higher reward of $(a_1', a_2', a_3')$ even though it is not a valid action. After this, the next two networks produce actions to ensure that the overall output is valid, e.g., the final action might be $(a_1', a_2'', a_3'')$---this action can be much worse than $(a_1^*, a_2^*, a_3^*)$.
> > > > >
> > > > > We will include a concise version of this discussion in the paper.

---

> > > > > > ### Comment · Reviewer_Xx2Q · 2023-08-16
> > > > > > **Response**
> > > > > >
> > > > > > That makes a lot of sense, thanks for the explanation!
> > > > > >
> > > > > > Putting some version of that in the paper seems valuable, yeah.
> > > > > >
> > > > > > With these additional comparisons and some clarifications to the text to avoid confusion, I'll raise my score here.

---

> > > > > > > ### Author Response · Authors · 2023-08-17
> > > > > > >
> > > > > > > Thank you for your positive feedback and for raising the score. We appreciate your time and effort in reviewing our paper.

---

> > > > > > > ### Author Response · Authors · 2023-08-17
> > > > > > >
> > > > > > > We additionally thank you for engaging with us to highlight and contrast the contributions of our work. These additional comparisons and clarifications, which we will add, would make the paper more comprehensive and understandable.

---

### Official Review · Reviewer_Ed9B · 2023-07-06

**Soundness:** 3 good
**Presentation:** 3 good
**Contribution:** 2 fair
**Rating:** 6
**Confidence:** 3

**Summary:**

The paper proposes a new approach to solve RL problems with large categorical action spaces and state-dependent constraints. The approach consists of two main components: a flow-based policy network that uses a discrete normalizing flow model to represent the stochastic policy, and an invalid action rejection method that leverages an oracle to filter out actions that violate the constraints. The paper derives a modified policy gradient estimator for the proposed method and shows its effectiveness on control tasks (Acrobot, Pistonball) and resource allocation task (ERA).

**Strengths:**

The idea of the combination of discrete normalizing flow is novel, and the way of invalid action rejection to address the challenges of state-dependent constraints is simple but reasonable. Besides, the paper provides a sound theoretical foundation (though I did not check it very carefully) and empirical evaluation for the proposed method, showing its advantages over existing baselines. It provides a clear definition of the problem and reasonable hypotheses. The proposed methods are well presented, and the experiments demonstrate the effectiveness of the proposed method. Overall, the conclusions are supported by empirical evidence and theoretical analysis. I believe this paper develop a reasonable method for solving large action space and resource allocation problem.

**Weaknesses:**

There are some limitations/weakness I want to discuss:

### Method:

The motivation is not clear. I can not get the advantage of using generative policies to address resource allocation problems. The introduction section directly proposes the discrete normalizing flow policy. Especially, MASK, without generative policies, also demonstrates good performance in the experiments. Besides, the theorem/proof is straightforward and intuitive.




### Experiments:

1. The experiment results are weak on ERA tasks. Though IAR-A2C slightly outperforms MASK in terms of the final performance. However, it's clear that MASK learns more efficient on ERA-v[2-5], as shown in Figure 5. MASK is actually a simple method that only considers the action constraints by adding an action probability masking. Though author presents the wall time of the training (Figure 6, which partly alleviates my concern), there are not many scenarios where you need to query to know about the action constraints. Thus I am wondering the necessity of using the proposed method than MASK.

2. For ablation study, the paper does not compare the proposed method with other generative models for RL, such as GANs or diffusion models, which could provide alternative solutions to normalizing flow policy.

3. The paper tries to tackle the problem of resource allocation, but only evaluate the proposed method on the customized environment. It can be further improved by conducting experiment on more widely used publich benchmarks.

**Questions:**


1. It is not clear why 'such constraints are easy to validate by a validity oracle given any allocation action, but are hard to represent as mathematical constraints' (line 35). Can you provide a more concrete example?

2. How does the set of actions sample from the flow policy (Sample actions from flow policy in Figure 1)? Is it obtained by multiple sampling from the distribution of policy outputs? If so, why not switch to a single time of sampling until getting a valid action?

3. What is constraints in Acrobot and Pistonball tasks as they seems like common RL tasks?



**Limitations:**

Refer to 'Weakness' part. I would like to present some minor limitations that do not affect the score:
1. "dependent constraints.." -> "dependent constraints."
2. Introduce the motivation for using Flow based Policy network in the introduction section.
3. Add more descriptions about the experimental setting in the experiment section, e.g., details about the action constraints.

---

> ### Author Rebuttal · Authors · 2023-08-10
>
> We thank the reviewer for the comments and feedback.
>
> * MASK comparison: The MASK approach can work when the mask can be specified easily. For constraints that vary by state (need a mask per state) and where the number of actions is large specifying the mask is costly - particularly, for our main problem of emergency resource allocation specifying such a mask per state is simply not possibly (apriori specification and storing of such mask would requires enormous storage space).
> * Other generative models: The paragraph starting line 134 discusses the reason for our choice of normalizing flow - GAN does not provide probability density estimates and diffusion models are much more computationally heavy than normalizing flow.
> * Benchmark: There is no widely accepted benchmark for heterogeneous resource allocation, we design our benchmark inspired by some prior works.
>    Prior works did not follow OpenAI Gym API [1, 2, 3], which requires decent efforts on implementation to match their APIs. Also, we encountered dependency errors when we attempted to use the environments.
>
>    [1] Vehicular Network Environment: GitHub - neardws/Game-Theoretic-Deep-Reinforcement-Learning
>
>    [2] Decentralized resource allocation for vehicle-to-vehicle: GitHub - Engineer1999/Double-Deep-Q-Learning-for-Resource-Allocation
>
>    [3] Decentralized resource allocation for 5G vehicle-to-vehicle: GitHub - gundoganalperen/DIRAL
>
>
> Response to questions:
> 1. Suppose there are $M$ areas and resources $R_1, R_2, …, R_N$  to be allocated ($ M \approx 1.5N$). A constraint can be that any subset of $N/2$ areas must have at least one area covered by at least one resource.
>
>    Now, if given an allocation, it is easy to check the number of areas that have no allocation. If this number is $\geq N/2$ then the constraint is violated (i.e., there is a subset of size N/2 with no allocation) and if this number is $< N/2$ then the constraint is satisfied (as any subset of N/2 areas will contain an allocated one). This is easy to check.
>
>    For a mathematical representation, let $x_{n, m} \in \\{0,1\\}$ be a binary variable indicating allocation of resource $n$ to location $m$. Simple allocation constraints make sure that a resource is only allocated once: $\sum_m  x_{n, m} = 1$ for all n.
>    To make sure that a subset of targets $i_1, \ldots, i_k$ has at least one area allocated, we can write $\sum_{n,k}  x_{n, i_k} \geq 1$
>    Then, to enforce the constraint, we need
>    $\sum_{n,k}  x_{n, i_k} \geq 1$ for all possible subsets $\\{i_1, \ldots, i_k\\} \subset \\{1, \ldots, M\\}$ with $k = N/2$
>    The number of such constraints are $M \choose N/2$, which is large.
>    (We note that in many cases simplification of constraints might be possible but that is done typically on a case by case basis)
>
>
> 2. The samples are obtained by calling the conditional normalizing flow policy multiple times using multiple samples from the noise z. We need both valid and invalid samples to estimate the terms in the modified policy gradient. Sampling till getting the first valid sample makes such estimation complicated and likely higher variance (as sometimes the first sample itself might be valid).
> 3. As stated in the paper (e.g., caption of Figure 4), Acrobot and pistonball are the unconstrained case experiments in our main paper. Further, we have experiments with a constrained version of pistonball in Appendix C that restricts the upward movement of pistons on the left side to ensure the ball continues rolling to the left.

---

> > ### Comment · Reviewer_Ed9B · 2023-08-15
> >
> > Thanks for your detailed response. I have read the response and other reviewers' comments carefully, and decided to maintain my score.

---

> > > ### Author Response · Authors · 2023-08-15
> > >
> > > Thank you for your positive feedback. We appreciate your time and effort in reviewing our paper.

---

### Official Review · Reviewer_We8u · 2023-07-06

**Soundness:** 3 good
**Presentation:** 2 fair
**Contribution:** 3 good
**Rating:** 7
**Confidence:** 3

**Summary:**

The submission investigates reinforcement learning in the context of large categorical action spaces. The submission proposes parameterizing the policy using a state conditional normalizing flow in these settings. The submission derives a rejection-sampling-based method for handling valid action constraints determined by a black-box oracle. The submission shows the performance of its approach in experiments.

---

I am disclosing here that I have not checked the proofs of the theorems.

**Strengths:**

The problem that the submission investigates is both important and underexplored. The submission's approach seems reasonable and principled.

**Weaknesses:**

The submissions writes the following:

> Another type of approach relies on being able to represent a joint probability distribution over multi-dimensional discrete actions using marginal probability distribution over each dimension (also called factored representation). Tang and Agrawal [27] apply this approach on discretized continuous control problems to decrease the learning complexity. Similarly, Delalleau et al. [7] assumes independence among dimensions to model each dimension independently. However, it is well-known in optimization and randomized allocation literature [4, 32] that dimension independence or marginal representation is not valid in the presence of constraints on the support of the probability distribution. Another type of approach [33, 31] converts the choice of multi-dimensional discrete action into a sequential choice of action across the dimensions at each time step (using a LSTM based policy network), where the choice of action in any dimension is conditioned on actions chosen for prior dimensions. This heuristic also produces a deterministic policy, thus, it is not suited for problems where the optimal policy is stochastic.

I have two criticisms/questions about this passage. First, I don't understand the sentence referencing "a deterministic policy". It is of course possible for autoregressive models to parameterize non-deterministic policies. Could the authors clarify exactly what is meant here? Second, while I think the submission's criticisms of these approaches is fair, it's point would be made more strongly if it showed that these flaws resulted in worse experimental performance. Sometimes, it is the case that a "stupid but easy" approach ends up performing well in practice. If the submission were to show that this is not the case for the environments it examines, it would strengthen its value.

**Questions:**

> Think of the things where a response from the author can change your opinion, clarify a confusion or address a limitation

I think adding baselines that use factored or autoregressive action spaces could cause me to increase my score. I will not decrease my score if these additional baselines perform well -- I think it would be valuable for the community to know in either case.


**Limitations:**

> Have the authors adequately addressed the limitations

Yes

---

> ### Author Rebuttal · Authors · 2023-08-10
>
> We thank the reviewer for the comments and feedback.
>
> We agree that that autoregressive model can also produce stochastic policies. We will modify this statement. We have implemented an autoregressive model (unfortunately, we did not find any publicly available code) and results are in the attached pdf; also present are results for the marginal/factored approach.
>
> Figure 1 in the attached PDF shows that our approach has similar performance as the factored/marginal and the autoregressive approach in the **unconstrained case**, which is as per expectation since the dimensions are independent and the factored approach assumes underlying independence across dimensions. The autoregressive approach assumes a particular dependence structure (based on ordering of dimensions) but is equivalent to marginal A2C when there is underlying independence across dimensions. Figure 3 in the PDF shows that the autoregressive approach does not perform well in the **constrained case**.
>
> We have additional experiment still running and hope to share these during the discussion phase

---

> > ### Comment · Reviewer_We8u · 2023-08-15
> > **Thanks for performing the additional experiments**
> >
> > I am satisfied by the author response and have raised my score.

---

> > > ### Author Response · Authors · 2023-08-15
> > >
> > > Thank you for your positive feedback and for raising the score. We appreciate your time and effort in reviewing our paper.

---

### Official Review · Reviewer_NSnS · 2023-07-07

**Soundness:** 3 good
**Presentation:** 2 fair
**Contribution:** 2 fair
**Rating:** 5
**Confidence:** 3

**Summary:**

This paper studies challenges in Reinforcement Learning (RL) problems that involve finding an optimal policy where the action space is categorical and large in which current RL approaches do not have a good performance. In addition, these problems often require validity constraints on the actions, which are difficult to express mathematically. To tackle these challenges, the authors propose using discrete normalizing flows conditioned on states to represent the stochastic policies compactly. They employ an invalid action rejection method and derive a modified policy gradient for updating the base policy. Lastly, they evaluate their method by comparing its performance with other RL benchmarks on several tasks in which they achieve state-of-the-art results.

**Strengths:**

Overall Strengths:
1. I personally find the high-level idea of looking into ways to improve the performance of RL policies invaluable.
2. The idea of representing stochastic policies for discrete action spaces with discrete normalizing flows is well-worth pursuing.
3. The overall performance of the proposed method is superior to the benchmarks shown in the experimental section, which shows the effectiveness of the approach.

**Weaknesses:**

Overall Weaknesses:
1. This paper emphasizes that standard categorical policies implemented using a Softmax layer require large resources as the size of the action space grows. They then motivate how their proposed Argmax Flow-based model is instead a compact representation of the policy. However, it is very vague to me if this claim is really true. I have clarified this point in the “Questions” section of the review.
2. In some parts of the paper, there are terms used that are not described until later on in the paper which makes the paper a little bit difficult to follow. I have clarified this point in the “Questions” section of the review.
3. The experimental setup is limited. I would personally be interested in experiments that show how the proposed method compares to others in terms of compactness (as it is one of the focuses in the write-up). Moreover, comparing against other state-of-the art algorithms and benchmark tasks would be invaluable. The current experimental results are promising, but I still wonder if they generalize across different algorithms and tasks.

**Questions:**

1. In figure 2, it is not clear what K is. It would be helpful if the caption was more descriptive.
2. From my understanding, since the argmax is going to give us the action, it must be applied to a layer with dimension equal to the number of actions. Is my understanding correct? If yes, how is this compact compared to the categorical policies implemented using a Softmax layer?
3. How does the computational power required by your method compare standard categorical policies for the same task? To me, it seems that you require far more computation to represent the policy as computing q and the density (requiring computing a Jacobian) is required with your method whereas it is not with categorical policies.
4. On page 4, you show how the state encoder works. Is this the standard way to condition the normalizing flow on some given vector? I would assume that traditionally, the noise is concatenated or summed with the condition.
5. For the gradient update on the return (shown in equation 7), have you considered using reparameterization of the policy (since it is now represented with normalizing flows) to avoid using the re-inforce trick? I believe the fact that you can reparameterize the policy is an advantage as it can lead to potentially smarter gradient updates.
6. Can you provide any experimental results demonstrating the compactness of your method?
7. In figure 5, why is your model slower to converge than the other benchmarks?
8. In your experiments, do you add the same state encoder used for your approach, to the other benchmark algorithms? I believe that if the state encoder has a positive effect on your method, it has to be equally considered in other benchmarks you are comparing to.

**Limitations:**

Overall, this paper successfully introduces Argmax Flows as a representation method for categorical policies, achieving state-of-the-art results within the provided tasks and algorithms, which is commendable. However, at this stage, I perceive the proposed method as an alternative to standard categorical policies that may offer better performance but are computationally more demanding. To enhance the paper, further refinement is required in two key areas. First, clarify the claims regarding the superiority of this method. If its strength lies in compactness, provide a detailed explanation and demonstrate this aspect through experiments. If performance is the main advantage, showcase its generalizability across different tasks and algorithms. Additionally, it's important to note that I maintain an open mind and may reconsider my evaluation based on a compelling rebuttal.

---

> ### Author Rebuttal · Authors · 2023-08-10
>
> We thank the reviewer for the comments and feedback.
>
> We have implemented an autoregressive model (unfortunately, we did not find any publicly available code) and results are in the attached pdf; also present are results for the marginal/factored approach.
>
> Figure 1 in the attached PDF shows that our approach has similar performance as the factored/marginal and the autoregressive approach in the **unconstrained case**, which is as per expectation since the dimensions are independent and the factored approach assumes underlying independence across dimensions. The autoregressive approach assumes a particular dependence structure (based on ordering of dimensions) but is equivalent to marginal A2C when there is underlying independence across dimensions.
> Figure 3 in the PDF shows that the autoregressive approach does not perform well in the **constrained case**.
>
> We have additional experiment still running and hope to share these during the discussion phase.
>
> Response to questions:
>
> 1. We will make the caption more descriptive. K here is the last of K invertible neural network layers used in normalizing flow (described in lines 150-151)
>
>
> 2. The argmax is applied to the last dimension of the continuous $v \in \mathbb{R}^{D \times M}$. As a example, suppose number of dimensions are D=2 with M=3 choices in each dimension and if the continuous output by standard normalizing flow is $v = [[ 0.4, 0.5, 0.9], [2, 0.7, 1]]$ then argmax(v, dim=1) gives [2, 0], that is, choice 2 in first dimension and choice 0 in second dimension. Note that the output size of the continuous part is $D \times M$ and not the number of $M^D$ actions possible. We do note that all this is from the previous work in Argmax flow (described in lines 82-94)
>
>
> 3. Yes, compared to marginal (or independent categorical) representation, our approach will consume more computational power as we have to maintain an additional posterior network. But, our approach does not require computing Jacobians; the log determinant that shows in line 11 in Alg 1 is efficiently computed by appropriate choice of the invertible function $f_{w,k}$. There are many choices for $f_{w,k}$ used in normalizing flow literature all of which ensure efficient computation of this log determinant, more details can be found in
> [​​Papamakarios et al. 2021] (second last paragraph of page 11).
>
>    We also discuss our choice of $f_{w,k}$ in Appendix B.2.
>
>    [​​Papamakarios et al. 2021] ​​Papamakarios, George, Eric Nalisnick, Danilo Jimenez Rezende, Shakir Mohamed, and Balaji Lakshminarayanan. "Normalizing flows for probabilistic modeling and inference." The Journal of Machine Learning Research 22, no. 1 (2021): 2617-2680.
>
> 4. Unlike other generative models, in normalizing flow typically the size of the noise input is the same as the size $D \times M$ of the output (due to the invertible layers used in between). Given this, we would have been forced to generate a state encoding of size $D \times M$, which then would impose a size restriction for baselines. Thus, we used this approach which has been used in other papers also [Ward et al. 2019].
>
>    [Ward et al. 2019] Ward, P. N., Smofsky, A., & Bose, A. J. (2019). Improving exploration in soft-actor-critic with normalizing flows policies. arXiv preprint arXiv:1906.02771.
>
>
> 5. We have not considered reparameterization, though we agree that this is another interesting way of getting gradients made possible by use of the generative model as a policy network. We aim to explore this in future pertaining to what advantages would this offer over and above the reinforced style gradient we use.
>
>
> 6. As stated in answer 2, the compactness is self evident in the size of outputs $D \times M$ that we have used relative to the $M^D$ size that the standard A2C would use, given the size of the action space $|\mathcal A| = M^D$.
>
>
> 7. Our approach appears slower in the number of iterations, but the same result in actual wall clock time (shown in Figure 6 in main paper and Figure A.3 in appendix) shows the much faster convergence in time. This is because MASK takes a lot of time per iteration.
>
> 8. Yes, we use the state encoding for all baselines.

---

> > ### Comment · Reviewer_NSnS · 2023-08-19
> >
> > Thank you for your effort in running additional experiments and answering my questions. I had been eagerly awaiting an update on the status of these experiments. Could you please confirm whether the uploaded PDF contains all the results from the additional experiments?
> >
> > Having reviewed your responses to my questions and considered the insights from fellow reviewers, I have chosen to modestly increase my original rating.
> >
> > I continue to believe that this work would benefit from further refinement, particularly in terms of its written presentation. Clarity in motivation is essential, and extending the experiments to include a comparison with relevant baselines would significantly enhance the paper's value. I acknowledge that some of these suggestions have already been raised by the reviewers and are intended for incorporation in the final version of the paper.
> >
> > Moreover, with the current setup, I believe it is a waste not to take advantage of reparameterization of the policy. This could lead to potentially better results compared to baselines and would increase the quality of the paper as well. But as this may not be the main promise of the work, I respect the authors' decision to defer this exploration to future works.

---

> > > ### Author Response · Authors · 2023-08-21
> > > **Follow-up Response to Reviewer NSnS (1/2)**
> > >
> > > Thank you for your positive comment and for raising the score. We do have experiments other than those in the pdf, but we are unable to update the PDF at this stage.  Sorry for the delay in this reply, as we were waiting for some experiments to complete.
> > >
> > > We will improve the clarity of our motivation by contrasting with other approaches in writing (in Introduction/Abstract itself) and extend the experiment section with the comparisons to additional approaches that we have shown in text in this discussion. We summarize all additional results till now below for you.
> > >
> > > **Results in PDF**: Our approach has shown similar performance as the autoregressive approach and the factored/marginal approach in the **unconstrained case**, shown in Figure 1 in the PDF and discussed in our rebuttal. Autoregressive (AR) approach does not perform well in the **constrained case** (Figure 3 in the PDF).
> > >
> > > **Additional results presented to reviewer Xx2Q**: We have explained concisely why AR does not work in constrained case as well as provided a full detailed example experiment supporting our claims. Please refer to our discussion with the reviewer Xx2Q for the full example and our analysis on why it does not work.
> > >
> > > **New results presented here**: The factored/marginal approach has similar performance as our approach when there exists a deterministic optimal policy, including in the constrained case. But the factored approach has worse (sometimes much worse) performance than ours when the optimal policy is stochastic, which we show by the experiment results below:
> > >
> > > Summary of new results: We investigate the case where the optimal policy is stochastic and this stochastic policy is better than any other deterministic policy. We show experimentally that the factored approach suffers due to its inherent independent dimension assumption and cannot find the optimal policy in this case, while our approach does find the optimal policy.
> > >
> > >
> > > As we stated in our rebuttal to reviewer Xx2Q, an optimal stochastic policy that is better than any other deterministic policy can occur in partial observation settings. We note that the partial observability is the practical challenge in resource allocation problems [Wu et al. 2016].
> > >
> > > Wu, F., Ramchurn, S. D., & Chen, X. (2016). Coordinating human-UAV teams in disaster response. In Proceedings of the 25th International Joint Conference on Artificial Intelligence (IJCAI) (pp. 524-530).
> > >
> > >
> > > For purposes of better exposition, here we give one example in the unconstrained case to show that the stochastic policy is the optimal one and outperforms any deterministic policy. We also experimentally show that the factored/marginal approach is not able to learn the stochastic optimal policy, because of the fundamental reason that the optimal stochastic policy here **cannot** be represented by the product of the marginal distributions.

---

> > > > ### Author Response · Authors · 2023-08-21
> > > > **Follow-up Response to Reviewer NSnS (2/2)**
> > > >
> > > > First, we describe our example set-up, which we build off of an example in  [Singh et al. 1994]. The underlying system has 2 states, 1a and 1b, and 4 actions, (0, 0) (0, 1) (1, 0) (1, 1) and a discount factor $\gamma$. The payoff and transition are defined as follows:
> > > >
> > > > - In any state, action (0, 0) and (1, 1) would lead to the state being unchanged with an immediate reward of $-R$.
> > > > - In state 1a, action (0, 1) leads to the state 1a (the same state) with a reward $-R$; action (1, 0) leads to the state 1b with a reward $R$.
> > > > - In state 1b, action (0, 1) leads to the state 1a with a reward $R$; action (1, 0) leads to the state 1b (the same state) with a reward $-R$.
> > > >
> > > > In other words, the agent obtain a positive reward if the state changes, otherwise obtaining a negative reward. Due to the partial observability, the agent only sees one observation, i.e., cannot distinguish between the two states.
> > > >
> > > > [Singh et al. 1994] Singh, Satinder P., Tommi Jaakkola, and Michael I. Jordan. "Learning without state-estimation in partially observable Markovian decision processes." In Machine Learning Proceedings 1994, pp. 284-292. Morgan Kaufmann, 1994.
> > > >
> > > > In this example, the best stochastic policy is probability 0.5 on (0, 1) and probability 0.5 on (1, 0) given the single observation, which gets a long term reward of 0, which is better than any deterministic policy. The deterministic policy playing (0,1) or (1,0) gets $\frac{(1-2\gamma)}{(1-\gamma)}R < 0$ (this inequality holds for any $\gamma>0.5$) and the policy playing (0,0) or (1,1) gets $-\frac{R}{(1-\gamma)} < 0$ (when $\gamma>0.5$). Note that a large $R$ can make the optimal stochastic policy arbitrarily better than any deterministic policy.
> > > >
> > > > Now, we discuss experiment results. In our implementation, the reward $R$ is set to be 10, $\gamma$ is 0.99 and the maximal episode length is 100. Shown below is the policy learned by our approach and the factored approach given the observation (note there is only one observation), shown as a distribution over 200 sampled actions (Table 1).
> > > >
> > > >
> > > > Table 1. Policy learned by our approach and factored approach.
> > > >  **Action** | **Num - Ours** | **Num - Factored**|**Dist - Ours**|**Dist - Factored**|
> > > > ---|---|---|---|---|
> > > >  (0, 0)| 0| 51|0.000|0.255
> > > >  (0, 1)| 98| 45|0.490|0.245
> > > >  (1, 0)| 101| 43|0.505|0.215
> > > >  (1, 1)| 1| 61|0.005|0.305
> > > >
> > > > We observe that our approach learns a stochastic policy very close to the exact stochastic optimal one, 0.5 probability on (0, 1) and 0.5 probability on (1, 0). In contrast, the factored approach cannot effectively learn the optimal policy (its best policy is close to the random initialized policy), which result in a huge loss in return as presented below. We evaluate the learned policy over 10 episodes to get Table 2:
> > > >
> > > > Table 2. Average episodic return of our approach and factored approach.
> > > >  **Ours** | **Factored**
> > > > ---|---
> > > >  22.00 +/- 67.20 | -564.00 +/- 39.80
> > > >
> > > > Our approach gets the return close to 0 (expected return of the optimal policy), while the factored approach gets very low return, with a very wide gap of about 550 compared to our approach.
> > > >
> > > > In addition to this example, we perform experiments on our ERA environment (3 targets with 2 resources) by introducing partial observation. We find that the optimal policy becomes stochastic. Similar to the above example, factored approach fails to learn the stochastic optimal policy and obtains a lower return.
> > > >
> > > > We evaluate the learned policy over 10 episodes (Table 3):
> > > >
> > > > Table 3. Average episodic return of our approach and factored approach.
> > > >  **Ours**| **Factored**
> > > > ---|---
> > > >  344.00 +/- 110.56 | 124.00 +/- 64.99

---

### Author Rebuttal · Authors · 2023-08-10

We thank all the reviewers for their comments and feedback. We use this space to attach the one-page pdf with extra experiments. We discuss the results in specific responses to the reviewers who asked for the experiments.

---

### Decision · Program_Chairs · 2023-09-21

**Decision:**

Accept (poster)

**Comment:**

The reviewers unanimously agreed on acceptance.